# EQUIVARIANT HYPERGRAPH DIFFUSION NEURAL OPERATORS

**Peihao Wang** [1]*, **Shenghao Yang** [3], **Yunyu Liu** [4], **Zhangyang Wang** [1], **Pan Li** [2,4]*

[1]University of Texas at Austin, [2]Georgia Tech, [3]University of Waterloo, [4]Purdue University
`{peihaowang,atlaswang}@utexas.edu`, `shenghao.yang@uwaterloo.ca`,
`liu3154@purdue.edu`, `panli@gatech.edu`

## ABSTRACT

Hypergraph neural networks (HNNs) using neural networks to encode hypergraphs provide a promising way to model higher-order relations in data and further solve relevant prediction tasks built upon such higher-order relations. However, higher-order relations in practice contain complex patterns and are often highly irregular. So, it is often challenging to design an HNN that suffices to express those relations while keeping computational efficiency. Inspired by hypergraph diffusion algorithms, this work proposes a new HNN architecture named ED-HNN, which provably approximates any continuous equivariant hypergraph diffusion operators that can model a wide range of higher-order relations. ED-HNN can be implemented efficiently by combining star expansions of hypergraphs with standard message passing neural networks. ED-HNN further shows great superiority in processing heterophilic hypergraphs and constructing deep models. We evaluate ED-HNN for node classification on nine real-world hypergraph datasets. ED-HNN uniformly outperforms the best baselines over these nine datasets and achieves more than 2%↑ in prediction accuracy over four datasets therein. Our code is available at: `https://github.com/Graph-COM/ED-HNN`.

## 1 INTRODUCTION

Machine learning on graphs has recently attracted great attention in the community due to the ubiquitous graph-structured data and the associated inference and prediction problems (Zhu, 2005; Hamilton, 2020; Nickel et al., 2015). Current works primarily focus on graphs which can model only pairwise relations in data. Emerging research has shown that higher-order relations that involve more than two entities often reveal more significant information in many applications (Benson et al., 2021; Schaub et al., 2021; Battiston et al., 2020; Lambiotte et al., 2019; Lee et al., 2021). For example, higher-order network motifs build the fundamental blocks of many real-world networks (Mangan & Alon, 2003; Benson et al., 2016; Tsourakakis et al., 2017; Li et al., 2017; Li & Milenkovic, 2017). Session-based (multi-step) behaviors often indicate the preferences of web users in more precise ways (Xia et al., 2021; Wang et al., 2020; 2021; 2022). To capture these higher-order relations, hypergraphs provide a dedicated mathematical abstraction (Berge, 1984). However, learning algorithms on hypergraphs are still far underdeveloped as opposed to those on graphs.

Recently, inspired by the success of graph neural networks (GNNs), researchers have started investigating hypergraph neural network models (HNNs) (Feng et al., 2019; Yadati et al., 2019; Dong et al., 2020; Huang & Yang, 2021; Bai et al., 2021; Arya et al., 2020). Compared with GNNs, designing HNNs is more challenging. First, as aforementioned, higher-order relations modeled by hyperedges could contain complex information. Second, hyperedges in real-world hypergraphs are often of large and irregular sizes. Therefore, how to effectively represent higher-order relations while efficiently processing those irregular hyperedges is the key challenge when to design HNNs.

In this work, inspired by the recently developed hypergraph diffusion algorithms (Li et al., 2020a; Liu et al., 2021b; Fountoulakis et al., 2021; Takai et al., 2020; Tudisco et al., 2021a) we design a novel HNN architecture that holds provable expressiveness to approximate a large class of hypergraph diffusion while keeping computational efficiency. Hypergraph diffusion is significant due to

---

*Correspondence to: Peihao Wang and Pan Li.

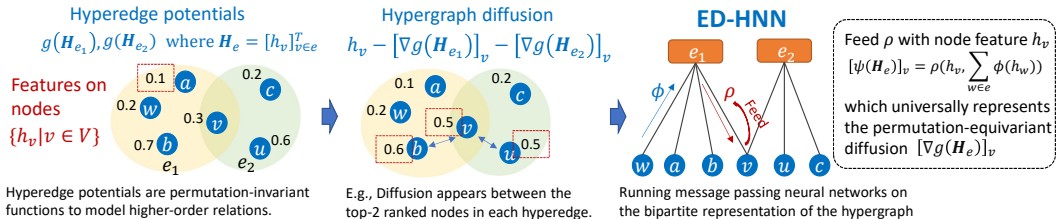

Figure 1: Hypergraph diffusion often uses permutation-invariant hyperedge potentials to model higher-order relations. The gradients of those potentials determine the diffusion process and are termed diffusion operators. Our ED-HNN can universally represent such operators by feeding node representations into the message from hyperedges to nodes. Such small changes make big differences in model performance.

its transparency and has been widely applied to semi-supervised learning (Hein et al., 2013; Zhang et al., 2017a; Tudisco et al., 2021a), ranking aggregation (Li & Milenkovic, 2017; Chitra & Raphael, 2019), network analysis (Liu et al., 2021b; Fountoulakis et al., 2021; Takai et al., 2020) and signal processing (Zhang et al., 2019; Schaub et al., 2021) and so on. However, traditional hypergraph diffusion needs to first handcraft potential functions to model higher-order relations and then use their gradients or some variants of their gradients as the diffusion operators to characterize the exchange of diffused quantities on the nodes within one hyperedge. The design of those potential functions often requires significant insights into the applications, which may not be available in practice.

We observe that the most commonly-used hyperedge potential functions are permutation invariant, which covers the applications where none of the nodes in a higher-order relation are treated as inherently special. For such potential functions, we further show that their induced diffusion operators must be permutation equivariant. Inspired by this observation, we propose a NN-parameterized architecture that is expressive to provably represent any *permutation-equivariant continuous hyperedge diffusion operators*, whose NN parameters can be learned in a data-driven way. We also introduce an efficient implementation based on current GNN platforms Fey & Lenssen (2019); Wang et al. (2019): We just need to combine a bipartite representation (or star expansion Agarwal et al. (2006); Zien et al. (1999), equivalently) of hypergraphs and the standard message passing neural network (MPNN) Gilmer et al. (2017). By repeating this architecture by layers with shared parameters, we finally obtain our model named Equivariant Diffusion-based HNN (ED-HNN). Fig. 1 shows an illustration of hypergraph diffusion and the key architecture in ED-HNN.

To the best of our knowledge, we are the first one to establish the connection between the general class of hypergraph diffusion algorithms and the design of HNNs. Previous HNNs were either less expressive to represent equivariant diffusion operators (Feng et al., 2019; Yadati et al., 2019; Dong et al., 2020; Huang & Yang, 2021; Chien et al., 2022; Bai et al., 2021) or needed to learn the representations by adding significantly many auxiliary nodes (Arya et al., 2020; Yadati, 2020; Yang et al., 2020). We provide detailed discussion of them in Sec. 3.4. We also show that due to the capability of representing equivariant diffusion operators, ED-HNN is by design good at predicting node labels over heterophilic hypergraphs where hyperedges mix nodes from different classes. Moreover, ED-HNN can go very deep without much performance decay.

As an extra theoretical contribution, our proof of expressiveness avoids using equivariant polynomials as a bridge, which allows precise representations of continuous equivariant set functions by compositing a continuous function and the sum of another continuous function on each set entry, while previous works (Zaheer et al., 2017; Segol & Lipman, 2020) have only achieved an approximation result. This result may be of independent interest for the community.

We evaluate ED-HNN by performing node classsification over 9 real-world datasets that cover both heterophilic and homophilic hypergraphs. ED-HNN uniformly outperforms all baseline methods across these datasets and achieves significant improvement ($>2\% \uparrow$) over 4 datasets therein. ED-HNN also shows super robustness when going deep. We also carefully design synthetic experiments to verify the expressiveness of ED-HNN to approximate pre-defined equivariant diffusion operators.

## 2 PRELIMINARIES: HYPERGRAPHS AND HYPERGRAPH DIFFUSION

Here, we formulate the hypergraph diffusion problem, along the way, introduce the notations.

**Definition 1** (Hypergraph). *Let $\mathcal{G} = (\mathcal{V}, \mathcal{E}, \boldsymbol{X})$ be an attributed hypergraph where $\mathcal{V}, \mathcal{E}$ are the node set and the hyperedge set, respectively. Each hyperedge $e = \{v_1^{(e)}, ..., v_{|e|}^{(e)}\}$ is a subset of $\mathcal{V}$. Unlike graphs, a hyperedge may contain more than two nodes. $\boldsymbol{X} = [..., \boldsymbol{x}_v, ...]^T \in \mathbb{R}^N$ denotes the node attributes and $\boldsymbol{x}_v$ denotes the attribute of node $v$. Define $d_v = |\{e \in \mathcal{E} : v \in e\}|$ as the degree of node $v$. Let $\boldsymbol{D}$, $\boldsymbol{D}_e$ denote the diagonal degree matrix for $v \in \mathcal{V}$ and the sub-matrix for $v \in e$.*

Here, we use 1-dim attributes for convenient discussion while our experiments often have multi-dim attributes. Learning algorithms will combine attributes and hypergraph structures into (latent) features defined as follows, which can be further used to make prediction for downstream tasks.

**Definition 2** (Latent features). *Let $\boldsymbol{h}_v \in \mathbb{R}$ denote the (latent) features of node $v \in \mathcal{V}$. The feature vector $\boldsymbol{H} = [..., \boldsymbol{h}_v, ...]_{v \in \mathcal{V}}^\top \in \mathbb{R}^N$ includes node features as entries. Further, collect the features into a hyperedge feature vector $\boldsymbol{H}_e = \begin{bmatrix} \boldsymbol{h}_{e,1} & \cdots & \boldsymbol{h}_{e,|e|} \end{bmatrix}^\top \in \mathbb{R}^{|e|}$, where $\boldsymbol{h}_{e,i}$ corresponds to the feature $\boldsymbol{h}_{v_i^{(e)}}$ for some node $v_i^{(e)} \in e$. For any $v \in e$, there is one corresponding index $i \in \{1, ..., |e|\}$. Later, we may use subscripts $(e, i)$ and $(e, v)$ interchangeably if they cause no confusion.*

A widely-used heuristic to generate features $\boldsymbol{H}$ is via hypergraph diffusion algorithms.

**Definition 3** (Hypergraph Diffusion). *Define node potential functions $f(\cdot; \boldsymbol{x}_v) : \mathbb{R} \to \mathbb{R}$ for $v \in \mathcal{V}$ and hyperedge potential functions $g_e(\cdot) : \mathbb{R}^{|e|} \to \mathbb{R}$ for each $e \in \mathcal{E}$. The hypergrah diffusion combines the node attributes and the hypergraph structure and asks to solve*

$$\min_{\boldsymbol{H}} \sum_{v \in \mathcal{V}} f(\boldsymbol{h}_v; \boldsymbol{x}_v) + \sum_{e \in \mathcal{E}} g_e(\boldsymbol{H}_e). \tag{1}$$

*In practice, $g_e$ is often shared across hyperedges of the same size. Later, we ignore the subscript $e$.*

The two potential functions are often designed via heuristics in traditional hypergraph diffusion literatures. Node potentials often correspond to some negative-log kernels of the latent features and the attributes. For example, $f(\boldsymbol{h}_v; \boldsymbol{x}_v)$ could be $(\boldsymbol{h}_v - \boldsymbol{x}_v)_2^2$ when to compute hypergraph PageRank diffusion (Li et al., 2020a; Takai et al., 2020). Hyperedge potentials are more significant and complex, as they need to model those higher-order relations between more than two objects, which makes hypergraph diffusion very different from graph diffusion. Here list a few examples.

**Example 1** (Hyperedge potentials). *Some practical $g(\boldsymbol{H}_e)$ may be chosen as follows.*

- *Clique Expansion (CE, hyperedges reduced to cliques) plus pairwise potentials (Zhou et al., 2007): $\sum_{u,v \in e}(\boldsymbol{h}_v - \boldsymbol{h}_u)_2^2$ or with degree normalization $\sum_{u,v \in e}(\frac{\boldsymbol{h}_v}{\sqrt{d_v}} - \frac{\boldsymbol{h}_u}{\sqrt{d_u}})_2^2 \ (\triangleq g(\boldsymbol{D}_e^{-1/2}\boldsymbol{H}_e))$.*

- *Divergence to the mean (Tudisco et al., 2021a;b): $\sum_{v \in e}(\boldsymbol{h}_v - \||e|^{-1}\boldsymbol{H}_e\|_p)_2^2$, where $\|\cdot\|_p$ computes the $\ell_p$-norm.*

- *Total Variation (TV) (Hein et al., 2013; Zhang et al., 2017a): $\max_{u,v \in e} |\boldsymbol{h}_v - \boldsymbol{h}_u|^p$, $p \in \{1, 2\}$.*

- *Lovász Extension (Lovász, 1983) for cardinality-based set functions (LEC) (Jegelka et al., 2013; Li et al., 2020a; Liu et al., 2021b)$\langle \boldsymbol{y}, \varsigma(\boldsymbol{H}_e)\rangle^p$, $p \in \{1, 2\}$ where $\boldsymbol{y} = [..., \boldsymbol{y}_j, ...]^\top \in \mathbb{R}^{|e|}$ is a constant vector and $\varsigma(\boldsymbol{H}_e)$ sorts the values of $\boldsymbol{H}_e$ in a decreasing order.*

One may reproduce TV by using LEC and setting $\boldsymbol{y}_1 = -\boldsymbol{y}_{|e|} = 1$. To reveal more properties of these hyperedge potentials $g(\cdot)$ later, we give the following definitions.

**Definition 4** (Permutation Invariance & Equivariance). *Function $\psi : \mathbb{R}^K \to \mathbb{R}$ is permutation-invariant if for any $K$-dim permutation matrix $\boldsymbol{P} \in \Pi[K]$, $\psi(\boldsymbol{P}\boldsymbol{Z}) = \psi(\boldsymbol{Z})$ for all $\boldsymbol{Z} \in \mathbb{R}^K$. Function $\psi : \mathbb{R}^K \to \mathbb{R}^K$ is permutation-equivariant if for any $K$-dim permutation matrix $\boldsymbol{P} \in \Pi[K]$, $\psi(\boldsymbol{P}\boldsymbol{Z}) = \boldsymbol{P}\psi(\boldsymbol{Z})$ for all $\boldsymbol{Z} \in \mathbb{R}^K$.*

We may easily verify permutation invariance of the hyperedge potentials in Example 1. The underlying physical meaning is that the prediction goal of an application is independent of node identities in a hyperedge so practical $g$'s often keep invariant w.r.t. the node ordering (Veldt et al., 2021).

## 3 Neural Equivariant Diffusion Operators and ED-HNN

Previous design of hyperedge potential functions is tricky. Early works adopted clique or star expansion by reducing hyperedges into edges (Zhou et al., 2007; Agarwal et al., 2006; Li & Milenkovic,

2017) and further used traditional graph methods. Later, researchers proved that those hyperedge reduction techniques cannot well represent higher-order relations (Li & Milenkovic, 2017; Chien et al., 2019). Therefore, Lovász extensions of set-based cut-cost functions on hyperedges have been proposed recently and used as the potential functions (Hein et al., 2013; Li & Milenkovic, 2018; Li et al., 2020a; Takai et al., 2020; Fountoulakis et al., 2021; Yoshida, 2019). However, designing those set-based cut costs is practically hard and needs a lot of trials and errors. Other types of handcrafted hyperedge potentials to model information propagation can also be found in (Neuhäuser et al., 2022; 2021), which again are handcrafted and heavily based on heuristics and evaluation performance.

Our idea uses data-driven approaches to model such potentials, which naturally brings us to HNNs. On one hand, we expect to leverage the extreme expressive power of NNs to learn the desired hypergraph diffusion automatically from the data. On the other hand, we are interested in having novel hypergraph NN (HNN) architectures inspired by traditional hypergraph diffusion solvers.

To achieve the goals, next, we show by the gradient descent algorithm (GD) or alternating direction method of multipliers (ADMM) (Boyd et al., 2011), solving objective Eq. 1 amounts to iteratively applying some hyperedge diffusion operators. Parameterizing such operators using NNs by each step can unfold hypergraph diffusion into an HNN. The roadmap is as follows.

In Sec. 3.1, we make the key observation that the diffusion operators are inherently *permutation-equivariant*. To universally represent them, in Sec. 3.2, we propose an equivariant NN-based operators and use it to build ED-HNN via an efficient implementation. In Sec. 3.3, we discuss the benefits of learning equivariant diffusion operators. In Sec. 3.4, we review and discuss that no previous HNNs allows efficiently modeling such equivariant diffusion operators via their architecture.

## 3.1 EMERGING EQUIVARIANCE IN HYPERGRAPH DIFFUSION

We start with discussing the traditional solvers for Eq. 1. If $f$ and $g$ are both differentiable, one straightforward optimization approach is to adopt gradient descent. The node-wise update of each iteration can be formulated as below:

$$\boldsymbol{h}_v^{(t+1)} \leftarrow \boldsymbol{h}_v^{(t)} - \eta(\nabla f(\boldsymbol{h}_v^{(t)}; \boldsymbol{x}_v) + \sum_{e:v\in e} [\nabla g(\boldsymbol{H}_e^{(t)})]_v), \quad \text{for } v \in \mathcal{V}, \tag{2}$$

where $[\nabla g(\boldsymbol{H}_e)]_v$ denotes the gradient w.r.t. $\boldsymbol{h}_v$ for $v \in e$. We use the subscript $t$ to denote the number of the current iteration, $\boldsymbol{h}_v^{(0)} = \boldsymbol{x}_v$ is the initial features, and $\eta$ is known as the step size.

For general $f$ and $g$, we may adopt ADMM: For each $e \in \mathcal{E}$, we introduce an auxiliary variable $\boldsymbol{Q}_e = \begin{bmatrix} \boldsymbol{q}_{e,1} & \cdots & \boldsymbol{q}_{e,|e|} \end{bmatrix}^\top \in \mathbb{R}^{|e|}$. We initialize $\boldsymbol{h}_v^{(0)} = \boldsymbol{x}_v$ and $\boldsymbol{Q}_e^{(0)} = \boldsymbol{H}_e^{(0)}$. And then, iterate

$$\boldsymbol{Q}_e^{(t+1)} \leftarrow \mathbf{prox}_{\eta g}(2\boldsymbol{H}_e^{(t)} - \boldsymbol{Q}_e^{(t)}) - \boldsymbol{H}_e^{(t)} + \boldsymbol{Q}_e^{(t)}, \quad \text{for } e \in \mathcal{E}, \tag{3}$$

$$\boldsymbol{h}_v^{(t+1)} \leftarrow \mathbf{prox}_{\eta f(\cdot; \boldsymbol{x}_v)/d_v}(\sum_{e:v\in e} \boldsymbol{q}_{e,v}^{(t+1)}/d_v), \quad \text{for } v \in \mathcal{V}, \tag{4}$$

where $\mathbf{prox}_\psi(\boldsymbol{h}) \triangleq \arg\min_{\boldsymbol{z}} \psi(\boldsymbol{z}) + \frac{1}{2}\|\boldsymbol{z} - \boldsymbol{h}\|_2^2$ is the proximal operator. The detailed derivation can be found in Appendix A. The iterations have convergence guarantee under closed convex assumptions of $f, g$ (Boyd et al., 2011). However, our model does not rely on the convergence, as our model just runs the iterations with a given number of steps (aka the number of layers in ED-HNN). The operator $\mathbf{prox}_\psi(\cdot)$ has nice properties as reviewed in Proposition 1, which enables the possibility of NN-based approximation even for the case when $f$ and $g$ are not differentiable. One noted non-differentiable example is the LEC case when $g(\boldsymbol{H}_e) = \langle \boldsymbol{y}, \varsigma(\boldsymbol{H}_e)\rangle^p$ in Example 1.

**Proposition 1** (Parikh & Boyd (2014); Polson et al. (2015)). *If $\psi(\cdot) : \mathbb{R}^K \rightarrow \mathbb{R}$ is a lower semi-continuous convex function, then $\mathbf{prox}_\psi(\cdot)$ is 1-Lipschitz continuous.*

The node-side operations, gradient $\nabla f(\cdot; \boldsymbol{x}_v)$ and proximal gradient $\mathbf{prox}_{\eta f(\cdot; \boldsymbol{x}_v)}(\cdot)$ are relatively easy to model, while the operations on hyperedges are more complicated. We name gradient $\nabla g(\cdot) : \mathbb{R}^{|e|} \rightarrow \mathbb{R}^{|e|}$ and proximal gradient $\mathbf{prox}_{\eta g}(\cdot) : \mathbb{R}^{|e|} \rightarrow \mathbb{R}^{|e|}$ as *hyperedge diffusion operators*, since they summarize the collection of node features inside a hyperedge and dispatch the aggregated information to interior nodes individually. Next, we reveal one crucial property of those hyperedge diffusion operators by the following propostion (see the proof in Appendix B.1):

**Proposition 2.** *Given any permutation-invariant hyperedge potential function $g(\cdot)$, hyperedge diffusion operators $\mathbf{prox}_{\eta g}(\cdot)$ and $\nabla g(\cdot)$ are permutation equivariant.*

---

**Algorithm 1: ED-HNN**

---

**Initialization:** $\boldsymbol{H}^{(0)} = \boldsymbol{X}$ **and three MLPs** $\hat{\phi}, \hat{\rho}, \hat{\varphi}$ **(shared across** $L$ **layers).**

**For** $t = 0, 1, 2, ..., L - 1$, **do:**

1. Designing the messages from $\mathcal{V}$ to $\mathcal{E}$: $\quad \boldsymbol{m}_{u \to e}^{(t)} = \hat{\phi}(\boldsymbol{h}_u^{(t)})$, for all $u \in \mathcal{V}$.

2. Sum $\mathcal{V} \to \mathcal{E}$ messages over hyperedges $\boldsymbol{m}_e^{(t)} = \sum_{u \in e} \boldsymbol{m}_{u \to e}^{(t)}$, for all $e \in \mathcal{E}$.

3. Broadcast $\boldsymbol{m}_e^{(t)}$ and design the messages from $\mathcal{E}$ to $\mathcal{V}$: $\boldsymbol{m}_{e \to v}^{(t)} = \hat{\rho}(\boldsymbol{h}_v^{(t)}, \boldsymbol{m}_e^{(t)})$, for all $v \in e$.

4. Update $\boldsymbol{h}_v^{(t+1)} = \hat{\varphi}(\boldsymbol{h}_v^{(t)}, \sum_{e:u \in e} \boldsymbol{m}_{e \to u}^{(t)}, \boldsymbol{x}_v, d_v)$, for all $v \in \mathcal{V}$.

---

It states that an permutation invariant hyperedge potential leads to an operator that should process different nodes in a permutation-equivariant way.

## 3.2 Building Equivariant Hyperedge Diffusion Operators

Our design of permutation-equivariant diffusion operators is built upon the following Theorem 1. We leave the proof in Appendix B.2.

**Theorem 1.** $\psi(\cdot) : [0, 1]^K \to \mathbb{R}^K$ *is a continuous permutation-equivariant function, if and only if it can be represented as* $[\psi(\boldsymbol{Z})]_i = \rho(\boldsymbol{z}_i, \sum_{j=1}^K \phi(\boldsymbol{z}_j))$, $i \in [K]$ *for any* $\boldsymbol{Z} = [..., \boldsymbol{z}_i, ...]^\top \in [0, 1]^K$, *where* $\rho : \mathbb{R}^{K'} \to \mathbb{R}$, $\phi : \mathbb{R} \to \mathbb{R}^{K'-1}$ *are two continuous functions, and* $K' \geq K$.

**Remark on an extra theoretical contribution:** Theorem 1 indicates that any continuous permutation-equivariant operators where each entry of the input $\boldsymbol{Z}$ has 1-dim feature channel can be precisely written as a composition of a continuous function $\rho$ and the sum of another continuous function $\phi$ on each input entry. This result generalizes the representation of permutation-invariant functions by $\rho(\sum_{i=1}^K \phi(\boldsymbol{z}_i))$ in (Zaheer et al., 2017) to the equivariant case. An architecture in a similar spirit was proposed in (Segol & Lipman, 2020). However, their proof only allows approximation, i.e., small $\|\psi(\boldsymbol{Z}) - \hat{\psi}(\boldsymbol{Z})\|$ instead of precise representation in Theorem 1. Also, Zaheer et al. (2017); Segol & Lipman (2020); Sannai et al. (2019) focus on representations of a single set instead of a hypergraph with coupled sets.

The above theoretical observation inspires the design of equivariant hyperedge diffusion operators. Specifically, for an operator $\hat{\psi}(\cdot) : \mathbb{R}^{|e|} \to \mathbb{R}^{|e|}$ that may denote either gradient $\nabla g(\cdot)$ or proximal gradient $\mathbf{prox}_{\eta g}(\cdot)$ for each hyperedge $e$, we may parameterize it as:

$$[\hat{\psi}(\boldsymbol{H}_e)]_v = \hat{\rho}(\boldsymbol{h}_v, \sum_{u \in e} \hat{\phi}(\boldsymbol{h}_u)), \text{ for } v \in e, \text{ where } \hat{\rho}, \hat{\phi} \text{ are multi-layer perceptions (MLPs).} \quad (5)$$

Intuitively, the inner sum collects the $\hat{\phi}$-encoding node features within a hyperedge and then $\hat{\rho}$ combines the collection with the features from each node further to perform separate operation.

The implementation of the above $\hat{\psi}$ is not trivial. A naive implementation is to generate an auxillary node to representation each $(v, e)$-pair for $v \in \mathcal{V}$ and $e \in \mathcal{E}$ and learn its representation as adopted in (Yadati, 2020; Arya et al., 2020; Yang et al., 2020). However, this may substantially increase the model complexity. Our implementation is built upon the bipartite representations (or star expansion (Zien et al., 1999; Agarwal et al., 2006), equivalently) of hypergraphs paired with the standard message passing NN (MPNN) (Gilmer et al., 2017) that can be efficiently implemented via GNN platforms (Fey & Lenssen, 2019; Wang et al., 2019) or sparse-matrix multiplication.

Specifically, we build a bipartite graph $\bar{\mathcal{G}} = (\bar{\mathcal{V}}, \bar{\mathcal{E}})$. The node set $\bar{\mathcal{V}}$ contains two parts $\mathcal{V} \cup \mathcal{V}_\mathcal{E}$ where $\mathcal{V}$ is the original node set while $\mathcal{V}_\mathcal{E}$ contains nodes that correspond to original hyperedges $e \in \mathcal{E}$. Then, add an edge between $v \in \mathcal{V}$ and $e \in \mathcal{E}$ if $v \in e$. With this bipartite graph representation, the model **ED-HNN** is implemented by following **Algorithm 1**.

The equivariant diffusion operator $\hat{\psi}$ can be constructed via steps 1-3. The last step is to update the node features to accomplish the first two terms in Eq. 2 or the ADMM update Eq. 4. We leave a more detailed discussion on how Algorithm 1 aligns with the GD and ADMM updates in Sec. C. The initial attributes $\boldsymbol{x}_v$ and node degrees are included to match the diffusion algorithm by design. As the diffusion operators are shared across iterations, ED-HNN shares parameters across layers. Now, we summarize the joint contribution of our theory and efficient implementation as follows.

**Proposition 3.** *MPNNs on bipartite representations (or star expansions) of hypergraphs are expressive enough to learn any continuous diffusion operators induced by invariant hyperedge potentials.*

**Simple while Significant Architecture Difference.** Some previous works (Dong et al., 2020; Huang & Yang, 2021; Chien et al., 2022) also apply GNNs over bipartite representations of hypergraphs to build their HNN models, which look similar to ED-HNN. However, ED-HNN has a simple but significant architecture difference: Step 3 in Algorithm 1 needs to adopt $h_v^{(t)}$ as one input to compute the message $m_{e \to v}^{(t)}$. This is a crucial step to guarantee that the hyperedge operation by combining steps 1-3 forms an equivariant operator from $\{h_v^{(t)}\}_{v \in e} \to \{m_{e \to v}^{(t)}\}_{v \in e}$. However, previous models often by default adopt $m_{e \to v}^{(t)} = m_e^{(t)}$, which leads to an invariant operator. Such a simple change is significant as it guarantees universal approximation of equivariant operators, which leads to many benefits as to be discussed in Sec. 3.3. Our experiments will verify these benefits.

**Extension.** Our ED-HNN can be naturally extended to build equivariant node operators because of the duality between hyperedges and nodes, though this is not necessary in the traditional hypergraph diffusion problem Eq. 1. Specifically, we call this model ED-HNNII, which simple revises the step 1 in ED-HNN as $m_{u \to e}^{(t)} = \hat{\phi}(h_u^{(t)}, m_e^{(t-1)})$ for any $e$ such that $u \in e$. Due to the page limit, we put some experiment results on ED-HNNII in Appendix F.1.

## 3.3 ADVANTAGES OF ED-HNN IN HETEROPHILIC SETTINGS AND DEEP MODELS

Here, we discuss several advantages of ED-HNN due to the design of equivariant diffusion operators.

Heterophily describes the network phenomenon where nodes with the same labels and attributes are less likely to connect to each other directly (Rogers, 2010). Predicting node labels in heterophilic networks is known to be more challenging than that in homophilic networks, and thus has recently become an important research direction (Pei et al., 2020; Zhu et al., 2020; Chien et al., 2021; Lim et al., 2021). Heterophily has been proved as a more common phenomenon in hypergraphs than in graphs since it is hard to expect all nodes in a giant hyperedge to share a common label (Veldt et al., 2022). Moreover, predicting node labels in heterophilic hypergraphs is more challenging than that in graphs as a hyperedge may consist of the nodes from multiple categories.

Learnable equivariant diffusion operators are expected to be superior in predicting heterophilic node labels. For example, if a hyperedge $e$ of size 3 is known to cover two nodes $v, u$ from class $\mathcal{C}_1$ while one node $w$ from class $\mathcal{C}_0$. We may use the LEC potential $\langle y, \varsigma(H_e) \rangle^2$ by setting the parameter $y = [1, -1, 0]^\top$. Suppose the three nodes' attributes are $H_e = [h_v, h_u, h_w]^\top = [0.7, 0.5, 0.3]^\top$, where $h_v, h_u$ are close as they are from the same class. One may check the hyperedge diffusion operator gives $\nabla g(H_e) = [0.4, -0.4, 0]^\top$. One-step gradient descent $h_v - \eta[\nabla g(H_e)]_v$ with a proper step size ($\eta < 0.5$) drags $h_v$ and $h_u$ closer while keeping $h_w$ unchanged. However, invariant diffusion by forcing the operation on hyperedge $e$ to follow $[\nabla g(H_e)]_v = [\nabla g(H_e)]_u = [\nabla g(H_e)]_w$ (invariant messages from $e$ to the nodes in $e$) will allocate every node with the same change and cannot deal with the heterogeneity of node labels in $e$. Moreover, a learnable operator is important. To see this, Suppose we have different ground-truth labels, say $w, u$ from the same class and $v$ from the other. Then, using the above parameter $y$ may increase the error. However, learnable operators can address the problem, e.g., by obtaining a more suitable parameter $y = [0, 1, -1]^\top$ via training.

Moreover, equivariant hyperedge diffusion operators are also good at building deep models. GNNs tend to degenerate the performance when going deep, which is often attributed to their oversmoothing (Li et al., 2018; Oono & Suzuki, 2019) and overfitting problems (Cong et al., 2021). Equivariant operators allocating different messages across nodes helps with overcoming the oversmoothing issue. Moreover, diffusion by sharing parameters across layers may reduce the risk of overfitting.

## 3.4 RELATED WORKS: PREVIOUS HNNS FOR REPRESENTING HYPERGRAPH DIFFUSION

ED-HNN is the first HNN inspired by the general class of hypergraph diffusion and can provably achieve universal approximation of hyperedge diffusion operators. So, how about previous HNNs representing hypergraph diffusion? We temporarily ignore the big difference in whether parameters are shared across layers and give some analysis as follows. HGNN (Feng et al., 2019) runs graph convolution (Kipf & Welling, 2017) on clique expansions of hypergraphs, which directly assumes the hyperedge potentials follow CE plus pairwise potentials and cannot learn other operations on hyperedges. HyperGCN (Yadati et al., 2019) essentially leverages the total variation potential by adding mediator nodes to each hyperedge, which adopts a transformation technique of the total variation potential in (Chan et al., 2018; Chan & Liang, 2020). GHSC (Zhang et al., 2022) is

Table 1: Dataset statistics. CE homophily is the homophily score (Pei et al., 2020) based on CE of hypergraphs. Here, four hypergraphs Congress, Senate, Walmart, House are heterophilic.

|  | Cora | Citeseer | Pubmed | Cora-CA | DBLP-CA | Congress | Senate | Walmart | House |
|---|---|---|---|---|---|---|---|---|---|
| # nodes | 2708 | 3312 | 19717 | 2708 | 41302 | 1718 | 282 | 88860 | 1290 |
| # edges | 1579 | 1079 | 7963 | 1072 | 22363 | 83105 | 315 | 69906 | 340 |
| # classes | 7 | 6 | 3 | 7 | 6 | 2 | 2 | 11 | 2 |
| avg. $|e|$ | 3.03 | 3.200 | 4.349 | 4.277 | 4.452 | 8.656 | 17.168 | 6.589 | 34.730 |
| CE Homophily | 0.897 | 0.893 | 0.952 | 0.803 | 0.869 | 0.555 | 0.498 | 0.530 | 0.509 |

to represent a specific edge-dependent-node-weight hypergraph diffusion proposed in (Chitra & Raphael, 2019). A few works view each hyperedge as a multi-set of nodes and each node as a multi-set of hyperedges (Dong et al., 2020; Huang & Yang, 2021; Chien et al., 2022; Bai et al., 2021; Arya et al., 2020; Yadati, 2020; Yang et al., 2020; Jo et al., 2021). Among them, HNHN (Dong et al., 2020), UniGNNs (Huang & Yang, 2021), EHGNN (Jo et al., 2021) and AllSet (Chien et al., 2022) build two invariant set-pooling functions on both the hyperedge and node sides, which cannot represent equivariant functions. HCHA (Bai et al., 2021) and UniGAT (Yang et al., 2020) compute attention weights as a result of combining node features and hyperedge messages, which looks like our $\hat{\rho}$ operation in step 3. However, a scalar attention weight is too limited to represent the potentially complicated $\hat{\rho}$. HyperSAGE (Arya et al., 2020), MPNN-R (Yadati, 2020) and LEGCN (Yang et al., 2020) may learn hyperedge equivariant operators in principle by adding auxillary nodes to represent node-hyperedge pairs as aforementioned. However, because of the extra complexity, these models are either too slow or need to reduce the sizes of parameters to fit in memory, which constrains their actual performance in practice. Of course, none of these works have mentioned any theoretical arguments on universal representation of equivariant hypergraph diffusion operators as ours.

One subtle point is that Uni{GIN,GraphSAGE,GCNII} (Huang & Yang, 2021) adopt a jump link in each layer to pass node features from the former layer directly to the next and may expect to use a complicated node-side invariant function $\hat{\psi}'(\boldsymbol{h}_v^{(t)}, \sum_{e:v\in e} \boldsymbol{m}_e^{(t)})$ to approximate our $\hat{\psi}(\boldsymbol{h}_v^{(t)}, \sum_{e:v\in e} \boldsymbol{m}_{e\to v}^{(t)}) = \hat{\psi}(\boldsymbol{h}_v^{(t)}, \sum_{e:v\in e} \hat{\rho}(\boldsymbol{h}_v^{(t)}, \boldsymbol{m}_e^{(t)}))$ in Step 4 of **Algorithm 1** directly. This may be doable in theory. However, the jump-link solution may expect to have a higher dimension of $\boldsymbol{m}_e^{(t)}$ (dim=$|\cup_{e:v\in e} e|$) so that the sum pooling $\sum_{e:v\in e} \boldsymbol{m}_e^{(t)}$ does not lose anything from the neighboring nodes of $v$ before interacting with $\boldsymbol{h}_v^{(t)}$, while in ED-HNN, $\boldsymbol{m}_e^{(t)}$ needs dim=$|e|$ according to Theorem 1. Our empirical experiments also verify more expressiveness of ED-HNN.

## 4 EXPERIMENTS

### 4.1 RESULTS ON BENCHMARKING DATASETS

**Experiment Setting.** In this subsection, we evaluate ED-HNN on nine real-world benchmarking hypergraphs. We focus on the semi-supervised node classification task. The nine datasets include co-citation networks (Cora, Citeseer, Pubmed), co-authorship networks (Cora-CA, DBLP-CA) Yadati et al. (2019), Walmart Amburg et al. (2020), House Chodrow et al. (2021), Congress and Senate Fowler (2006b;a). More details of these datasets can be found in Appendix F.2. Since the last four hypergraphs do not contain node attributes, we follow the method of Chien et al. (2022) to generate node features from label-dependent Gaussian distribution Deshpande et al. (2018).

As we show in Table 1, these datasets already cover sufficiently diverse hypergraphs in terms of scale, structure, and homo-/heterophily. We compare our method with top-performing models on these benchmarks, including HGNN (Feng et al., 2019), HCHA (Bai et al., 2021), HNHN (Dong et al., 2020), HyperGCN (Yadati et al., 2019), UniGCNII (Huang & Yang, 2021), AllDeepSets (Chien et al., 2022), AllSetTransformer (Chien et al., 2022), and a recent diffusion method HyperND (Prokopchik et al., 2022). All the hyperparameters for baselines follow from (Chien et al., 2022) and we fix the learning rate, weight decay and other training recipes same with the baselines. Other model specific hyperparameters are obtained via grid search (see Appendix F.3). We randomly split the data into training/validation/test samples using 50%/25%/25% splitting percentage by following Chien et al. (2022). We choose prediction accuracy as the evaluation metric. We run each model for ten times with different training/validation splits to obtain the standard deviation. In Appendix F.7, we also provide the analysis of the model sensitivity to different hidden dimensions.

**Performance Analysis.** Table 2 shows the results. Our ED-HNN uniformly outperforms all the compared models on all the datasets. We observe that the top-performing baseline models are AllSetTransformer, AllDeepSets and UniGCNII. As having been analyzed in Sec. 3.4, they model

Table 2: Prediction Accuracy (%). **Bold font**[†] highlights when ED-HNN significantly (difference in means $> 0.5 \times$ std) outperforms all baselines. The underlined best baselines are underlined. The training and testing times are test on Walmart by using the same server with one GPU NVIDIA RTX A6000.

| Homophilic Hypergraphs | Cora | Citeseer(Homo.) | Pubmed | Cora-CA | DBLP-CA | Training Time ($10^{-1}$ s) |
|---|---|---|---|---|---|---|
| HGNN Huang & Yang (2021) | 79.39 ± 1.36 | 72.45 ± 1.16 | 86.44 ± 0.44 | 82.64 ± 1.65 | 91.03 ± 0.20 | 0.24 ± 0.51 |
| HCHA Bai et al. (2021) | 79.14 ± 1.02 | 72.42 ± 1.42 | 86.41 ± 0.36 | 82.55 ± 0.97 | 90.92 ± 0.22 | 0.24 ± 0.01 |
| HNHN Dong et al. (2020) | 76.36 ± 1.92 | 72.64 ± 1.57 | 86.90 ± 0.30 | 77.19 ± 1.49 | 86.78 ± 0.29 | 0.30 ± 0.56 |
| HyperGCN Yadati et al. (2019) | 78.45 ± 1.26 | 71.28 ± 0.82 | 82.84 ± 8.67 | 79.48 ± 2.08 | 89.38 ± 0.25 | 0.42 ± 1.51 |
| UniGCNII Huang & Yang (2021) | 78.81 ± 1.05 | 73.05 ± 2.21 | 88.25 ± 0.40 | 83.60 ± 1.14 | 91.69 ± 0.19 | 4.36 ± 1.18 |
| HyperND Tudisco et al. (2021a) | 79.20 ± 1.14 | 72.62 ± 1.49 | 86.68 ± 0.43 | 80.62 ± 1.32 | 90.35 ± 0.26 | 0.15 ± 1.32 |
| AllDeepSets Chien et al. (2022) | 76.88 ± 1.80 | 70.83 ± 1.63 | 88.75 ± 0.33 | 81.97 ± 1.50 | 91.27 ± 0.27 | 1.23 ± 1.09 |
| AllSetTransformer Chien et al. (2022) | 78.58 ± 1.47 | 73.08 ± 1.20 | 88.72 ± 0.37 | 83.63 ± 1.47 | 91.53 ± 0.23 | 1.64 ± 1.63 |
| ED-HNN (ours) | **80.31 ± 1.35**[†] | **73.70 ± 1.38**[†] | **89.03 ± 0.53**[†] | 83.97 ± 1.55 | **91.90 ± 0.19**[†] | 1.71 ± 1.13 |
| Heterophilic Hypergraphs | Congress | Senate | Walmart | House | Avg. Rank | Inference Time ($10^{-2}$ s) |
| HGNN (Huang & Yang, 2021) | 91.26 ± 1.15 | 48.59 ± 4.52 | 62.00 ± 0.24 | 61.39 ± 2.96 | 5.22 | 1.01 ± 0.04 |
| HCHA (Bai et al., 2021) | 90.43 ± 1.20 | 48.62 ± 4.41 | 62.35 ± 0.26 | 61.36 ± 2.53 | 5.89 | 1.54 ± 0.18 |
| HNHN (Dong et al., 2020) | 53.35 ± 1.45 | 50.93 ± 6.33 | 47.18 ± 0.35 | 67.80 ± 2.59 | 6.67 | 6.11 ± 0.05 |
| HyperGCN (Yadati et al., 2019) | 55.12 ± 1.96 | 42.45 ± 3.67 | 44.74 ± 2.81 | 48.32 ± 2.93 | 8.22 | 0.87 ± 0.06 |
| UniGCNII (Huang & Yang, 2021) | 94.81 ± 0.81 | 49.30 ± 4.25 | 54.45 ± 0.37 | 67.25 ± 2.57 | 3.89 | 21.22 ± 0.13 |
| HyperND (Tudisco et al., 2021a) | 74.63 ± 3.62 | 52.82 ± 3.20 | 38.10 ± 3.86 | 51.70 ± 3.37 | 6.00 | 0.02 ± 0.01 |
| AllDeepSets (Chien et al., 2022) | 91.80 ± 1.53 | 48.17 ± 5.67 | 64.55 ± 0.33 | 67.82 ± 2.40 | 5.22 | 5.35 ± 0.33 |
| AllSetTransformer (Chien et al., 2022) | 92.16 ± 1.05 | 51.83 ± 5.22 | 65.46 ± 0.25 | 69.33 ± 2.20 | 2.88 | 6.06 ± 0.67 |
| ED-HNN (ours) | 95.00 ± 0.99 | **64.79 ± 5.14**[†] | **66.91 ± 0.41**[†] | **72.45 ± 2.28**[†] | 1.00 | 5.87 ± 0.36 |

Table 3: Prediction Accuracy (%) over Synthetic Hypergraphs with Controlled Heterophily $\alpha$.

| | Homophily | | | Heterophily | | |
|---|---|---|---|---|---|---|
| | $\alpha = 1$ | $\alpha = 2$ | $\alpha = 3$ | $\alpha = 4$ | $\alpha = 6$ | $\alpha = 7$ |
| HGNN (Huang & Yang, 2021) | 99.86 ± 0.05 | 90.42 ± 1.14 | 66.60 ± 1.60 | 57.90 ± 1.43 | 50.77 ± 1.94 | 48.68 ± 1.21 |
| HGNN + JumpLink (Huang & Yang, 2021) | 99.29 ± 0.19 | 92.61 ± 0.97 | 79.59 ± 2.32 | 64.39 ± 1.55 | 55.39 ± 2.01 | 51.14 ± 2.05 |
| AllDeepSets (Chien et al., 2022) | 99.86 ± 0.07 | 99.13 ± 0.31 | 93.52 ± 1.31 | 83.45 ± 1.42 | 65.38 ± 1.27 | 70.94 ± 0.14 |
| AllDeepSets + JumpLink (Chien et al., 2022) | 99.95 ± 0.07 | 99.08 ± 0.24 | 96.61 ± 1.45 | 86.45 ± 1.30 | 74.08 ± 1.02 | 71.52 ± 0.94 |
| AllSetTransformer (Chien et al., 2022) | 99.99 ± 0.01 | 99.81 ± 0.02 | 98.38 ± 0.81 | 96.83 ± 0.03 | 78.25 ± 0.16 | 74.68 ± 0.09 |
| ED-HNN (ours) | 99.96 ± 0.01 | **99.87 ± 0.02**[†] | **98.85 ± 0.25**[†] | **98.45 ± 0.23**[†] | **80.48 ± 0.70**[†] | **79.71 ± 0.27**[†] |

invariant set functions on both node and hyperedge sides. UniGCNII also adds initial and jump links, which accidentally resonates with our design principle (the step 4 in Algorithm 1). However, their performance has large variation across different datasets. For example, UniGCNII attains promising performance on citation networks, however, has subpar results on Walmart dataset. In contrast, our model achieves stably superior results, surpassing AllSet models by 12.9% on Senate and UniGCNII by 12.5% on Walmart. We owe our empirical significance to the theoretical design of exact equivariant function representation. Compared with HyperND, the SOTA hypergraph diffusion algorithm, our ED-HNN achieves superior performance on every dataset. HyperND provably reaches convergence to the "divergence to the mean" hyperedge potential (see Example 1), which gives it good performance on homophilic networks while only subpar performance on heterophilic datasets. Regarding the computational efficiency, we report the wall-clock times for training (for 100 epochs) and testing of all models over the largest hypergraph Walmart, where all models use the same hyperparameters that achieve the reported performance on the left. Our model achieves efficiency comparable to AllSet models (Chien et al., 2022) and is much faster than UniGCNII. So, the implementation of equivariant computation in ED-HNN is still efficient.

## 4.2 RESULTS ON SYNTHETIC HETEROPHILIC HYPERGRAPH DATASET

**Experiment Setting.** As discussed, ED-HNN is expected to perform well on heterophilic hypergraphs. We evaluate this point by using synthetic datasets with controlled heterophily. We generate data by using contextual hypergraph stochastic block model (Deshpande et al., 2018; Ghoshdastidar & Dukkipati, 2014; Lin & Wang, 2018). Specifically, we draw two classes of $2,500$ nodes each and then randomly sample 1,000 hyperedges. Each hyperedge consists of 15 nodes, among which $\alpha_i$ many are sampled from class $i$. We use $\alpha = \min\{\alpha_1, \alpha_2\}$ to denote the heterophily level. Afterwards, we generate label-dependent Gaussian node features with standard deviation 1.0. We test both homophilic ($\alpha = 1, 2$ or CE homophily$\geq 0.7$) and heterophilic ($\alpha = 4 \sim 7$ or CE homophily $\leq 0.7$) cases. We compare ED-HNN with HGNN, AllSet models and their variants with jump links. We follow previous 50%/25%/25% data splitting methods and repeated 10 times the experiment.

**Results.** Table 3 shows the results. On homophilic datasets, all the models can achieve good results, while ED-HNN keeps slightly better than others. Once $\alpha$ surpasses 3, i.e., entering the heterophilic regime, the superiority of ED-HNN is more obvious. The jump-link trick indeed also helps, while building equivariance as ED-HNN does directly provides more significant improvement.

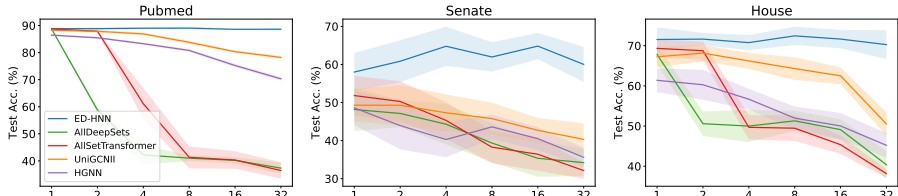

Figure 2: Comparing Deep Achitectures across Different Models: Test Acc. v.s. # Layers

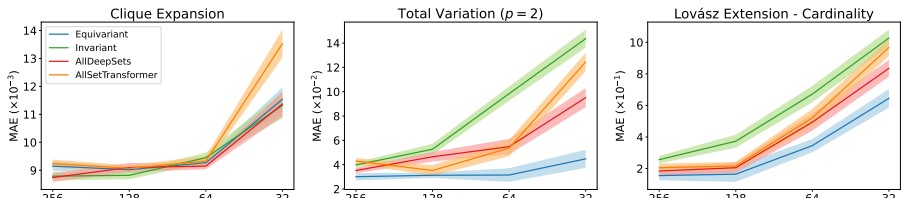

Figure 3: Comparing the Powers to Represent known Diffusion: MAE v.s. Latent Dimensions.

### 4.3 BENEFITS IN DEEPENING HYPERGRAPH NEURAL NETWORKS

We also demonstrate that by using diffusion models and parameter tying, ED-HNN can benefit from deeper architectures, while other HNNs cannot. Fig. 2 illustrates the performance of different models versus the number of network layers. We compare with HGNN, AllSet models , and UniGCNII. UniGCNII inherits from (Chen et al., 2020) which is known to be effective to counteract over-smoothness. The results reveal that AllSet models suffer from going deep. HGNN working through more lightweight mechanism has better tolerance to depth. However, none of them can benefit from deepening. On the contrary, ED-HNN successfully leverages deeper architecture to achieve higher accuracy. For example, adding more layers boosts ED-HNN by ~1% in accuracy on Pubmed and House, while elevating ED-HNN from 58.01% to 64.79% on Senate dataset.

### 4.4 EXPRESSIVENESS JUSTIFICATION ON THE SYNTHETIC DIFFUSION DATASET

We are to evaluate the ability of ED-HNN to express given hypergraph diffusion. We generate semi-synthetic diffusion data using the Senate hypergraph (Chodrow et al., 2021) and synthetic node features. The data consists of 1,000 pairs $(\boldsymbol{H}^{(0)}, \boldsymbol{H}^{(1)})$. The initial node features $\boldsymbol{H}^{(0)}$ are sampled from 1-dim Gaussian distributions. To obtain $\boldsymbol{H}^{(1)}$ we apply the gradient step in Eq. 2. For non-differential cases, we adopt subgradients for convenient computation. We fix node potentials as $f(\boldsymbol{h}_v; \boldsymbol{x}_v) = (\boldsymbol{h}_v - \boldsymbol{x}_v)^2$ and consider 3 different edge potentials in Example 1 with varying complexities: a) CE, b) TV ($p$=2) and c) LEC. The goal is to let one-layer models $\mathcal{V} \rightarrow \mathcal{E} \rightarrow \mathcal{V}$ to recover $\boldsymbol{H}^{(1)}$. We compare ED-HNN with our implemented baseline (Invariant) with parameterized invariant set functions on both the node and hyperedge sides, and AllSet models (Chien et al., 2022) that also adopt invariant set functions. We keep the scale of all models almost the same to give fair comparison, of which more details are given in Appendix F.5. The results are reported in Fig. 3. The CE case gives invariant diffusion so all models can learn it well. The TV case mostly shows the benefit of the equivariant architecture, where the error almost does not increase even when the dimension decreases to 32. The LEC case is challenging for all models, though ED-HNN is still the best. The reason, we think, is that learning the sorting operation in LEC via the sum pooling in Eq. 5 is empirically challenging albeit theoretically doable. A similar phenomenon has been observed in previous literatures (Murphy et al., 2018; Wagstaff et al., 2019).

## 5 CONCLUSION

This work introduces a new hypergraph neural network ED-HNN that can model hypergraph diffusion process. We show that any hypergraph diffusion with permutation-invariant potential functions can be represented by iterating equivariant diffusion operators. ED-HNN provides an efficient way to model such operators based on the commonly-used GNN platforms. ED-HNN shows superiority in processing heterophilic hypergraphs and constructing deep models. For future works, ED-HNN can be applied to other tasks such as regression tasks or to develop counterpart implicit models.

ACKNOWLEDGMENTS

We would like to express our deepest appreciation to Dr. David Gleich, Dr. Kimon Fountoulakis for the insightful discussion on hypergraph computation, and Dr. Eli Chien for the constructive advice on doing the experiments.

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

## A  DERIVATION OF EQ. 3 AND 4

We derive iterative update Eq. 3 and 4 as follows. Recall that our problem is defined as:

$$\min_{\boldsymbol{H}} \sum_{v \in \mathcal{V}} f(\boldsymbol{h}_v; \boldsymbol{x}_v) + \sum_{e \in \mathcal{E}} g_e(\boldsymbol{H}_e)$$

where $\boldsymbol{H} \in \mathbb{R}^N$ is the node feature matrix. $\boldsymbol{H}_e = \begin{bmatrix} \boldsymbol{h}_{e,1} & \cdots & \boldsymbol{h}_{e,|e|} \end{bmatrix}^\top \in \mathbb{R}^{|e|}$ collects the associated node features inside edge $e$, where $\boldsymbol{h}_{e,i}$ corresponds to the node features of the $i$-th node in edge $e$. We use $\boldsymbol{x}_v$ to denote the (initial) attribute of node $v$. Here, neither $f(\cdot; \boldsymbol{x}_v)$ nor $g_e(\cdot)$ is necessarily continuous. To solve such problem, we borrow the idea from ADMM (Boyd et al., 2011). We introduce an auxiliary variable $\boldsymbol{R}_e = \boldsymbol{H}_e$ for every $e \in \mathcal{E}$ and reformulate the problem as a constrained optimization problem:

$$\min_{\boldsymbol{H}} \sum_{v \in \mathcal{V}} f(\boldsymbol{h}_v; \boldsymbol{x}_v) + \sum_{e \in \mathcal{E}} g_e(\boldsymbol{R}_e)$$

$$\text{subject to } \boldsymbol{R}_e - \boldsymbol{H}_e = \boldsymbol{0}, \forall e \in \mathcal{E}$$

By the Augmented Lagrangian Method (ALM), we assign a Lagrangian multiplier $\boldsymbol{S}_e$ (scaled by $1/\lambda$) for each edge, then the objective function becomes:

$$\max_{\{\boldsymbol{S}_e\}_{e \in \mathcal{E}}} \min_{\boldsymbol{H}, \{\boldsymbol{R}_e\}_{e \in \mathcal{E}}} \sum_{v \in \mathcal{V}} f(\boldsymbol{h}_v; \boldsymbol{x}_v) + \sum_{e \in \mathcal{E}} g(\boldsymbol{R}_e) + \frac{\lambda}{2} \sum_{e \in \mathcal{E}} \|\boldsymbol{R}_e - \boldsymbol{H}_e + \boldsymbol{S}_e\|_F^2 - \frac{\lambda}{2} \sum_{e \in \mathcal{E}} \|\boldsymbol{S}_e\|_F^2. \quad (6)$$

We can iterate the following primal-dual steps to optimize Eq. 6. The primal step can be computed in a block-wise sense:

$$\boldsymbol{R}_e^{(t+1)} \leftarrow \arg\min_{\boldsymbol{R}_e} \left\{ \frac{\lambda}{2} \left\| \boldsymbol{R}_e - \boldsymbol{H}_e^{(t)} + \boldsymbol{S}_e^{(t)} \right\|_F^2 + g(\boldsymbol{R}_e) \right\}$$

$$= \mathbf{prox}_{g/\lambda} \left( \boldsymbol{H}_e^{(t)} - \boldsymbol{S}_e^{(t)} \right), \forall e \in \mathcal{E}, \quad (7)$$

$$\boldsymbol{h}_v^{(t+1)} \leftarrow \arg\min_{\boldsymbol{h}_v} \left\{ \frac{\lambda}{2} \sum_{e:v \in e} \left\| \boldsymbol{h}_v - \boldsymbol{S}_{e,v}^{(t)} - \boldsymbol{R}_{e,v}^{(t+1)} \right\|_F^2 + f(\boldsymbol{h}_v; \boldsymbol{x}_v) \right\}$$

$$= \mathbf{prox}_{f(\cdot; \boldsymbol{x}_v)/\lambda d_v} \left( \frac{\sum_{e:v \in e} (\boldsymbol{S}_{e,v}^{(t)} + \boldsymbol{R}_{e,v}^{(t+1)})}{d_v} \right), \forall v \in \mathcal{V}. \quad (8)$$

The dual step can be computed as:

$$\boldsymbol{S}_e^{(t+1)} \leftarrow \boldsymbol{S}_e^{(t)} + \boldsymbol{R}_e^{(t+1)} - \boldsymbol{H}_e^{(t+1)}, \forall e \in \mathcal{E}. \quad (9)$$

Denote $\boldsymbol{Q}_e^{(t+1)} := \boldsymbol{S}_e^{(t)} + \boldsymbol{R}_e^{(t+1)}$ and $\eta := 1/\lambda$, then the iterative updates become:

$$\boldsymbol{Q}_e^{(t+1)} = \mathbf{prox}_{\eta g}(2\boldsymbol{H}_e^{(t)} - \boldsymbol{Q}_e^{(t)}) + \boldsymbol{Q}_e^{(t)} - \boldsymbol{H}_e^{(t)}, \text{ for } e \in \mathcal{E}, \quad (10)$$

$$\boldsymbol{h}_v^{(t+1)} = \mathbf{prox}_{\eta f(\cdot; \boldsymbol{x}_v)/d_v} \left( \frac{\sum_{e:e \in v} \boldsymbol{Q}_{e,v}^{(t+1)}}{d_v} \right), \text{ for } v \in \mathcal{V}. \quad (11)$$

## B  DEFERRED PROOFS

### B.1  PROOF OF PROPOSITION 2

*Proof.* Define $\pi : [K] \to [K]$ be an index mapping associated with the permutation matrix $\boldsymbol{P} \in \Pi(K)$ such that $\boldsymbol{PZ} = \begin{bmatrix} \boldsymbol{z}_{\pi(1)}, \cdots, \boldsymbol{z}_{\pi(K)} \end{bmatrix}^\top$. To prove that $\nabla g(\cdot)$ is permutation equivariant, we show by using the definition of partial derivatives. For any $\boldsymbol{Z} = \begin{bmatrix} \boldsymbol{z}_1 & \cdots & \boldsymbol{z}_K \end{bmatrix}^\top$, and permutation $\pi$, we have:

$$[\nabla g(\boldsymbol{PZ})]_i = \lim_{\delta \to 0} \frac{g(\boldsymbol{z}_{\pi(1)}, \cdots, \boldsymbol{z}_{\pi(i)} + \delta, \cdots, \boldsymbol{z}_{\pi(K)}) - g(\boldsymbol{z}_{\pi(1)}, \cdots, \boldsymbol{z}_{\pi(i)}, \cdots, \boldsymbol{z}_{\pi(K)})}{\delta}$$

$$= \lim_{\delta \to 0} \frac{g(\boldsymbol{z}_1, \cdots, \boldsymbol{z}_{\pi(i)} + \delta, \cdots, \boldsymbol{z}_K) - g(\boldsymbol{z}_1, \cdots, \boldsymbol{z}_{\pi(i)}, \cdots, \boldsymbol{z}_K)}{\delta}$$

$$= [\nabla g(\boldsymbol{Z})]_{\pi(i)}, \quad (12)$$

where the second equality Eq. 12 is due to the permutation invariance of $g(\cdot)$. To prove Proposition 2 for the proximal gradient, we first define: $\boldsymbol{H}^* = \mathbf{prox}_g(\boldsymbol{Z}) = \arg\min_{\boldsymbol{H}} g(\boldsymbol{H}) + \frac{1}{2}\|\boldsymbol{H} - \boldsymbol{Z}\|_F^2$ for some $\boldsymbol{Z}$. For arbitrary permutation matrix $\boldsymbol{P} \in \Pi(K)$, we have

$$
\begin{aligned}
\mathbf{prox}_g(\boldsymbol{PZ}) &= \arg\min_{\boldsymbol{H}} g(\boldsymbol{H}) + \frac{1}{2}\|\boldsymbol{H} - \boldsymbol{PZ}\|_F^2 = \arg\min_{\boldsymbol{H}} g(\boldsymbol{H}) + \frac{1}{2}\|\boldsymbol{P}(\boldsymbol{P}^\top\boldsymbol{H} - \boldsymbol{Z})\|_F^2 \\
&= \arg\min_{\boldsymbol{H}} g(\boldsymbol{H}) + \frac{1}{2}\|\boldsymbol{P}^\top\boldsymbol{H} - \boldsymbol{Z}\|_F^2 = \arg\min_{\boldsymbol{H}} g(\boldsymbol{P}^\top\boldsymbol{H}) + \frac{1}{2}\|\boldsymbol{P}^\top\boldsymbol{H} - \boldsymbol{Z}\|_F^2 \\
&= \boldsymbol{PH}^*,
\end{aligned}
\tag{13}
$$

where Eq. 13 is due to the permutation invariance of $g(\cdot)$. $\square$

## B.2 PROOF OF THEOREM 1

*Proof.* To prove Theorem 1, we first summarize one of our key results in the following Lemma 2.

**Lemma 2.** $\psi(\cdot) : [0,1]^K \to \mathbb{R}^K$ *is a permutation-equivariant function if and only if there is a function* $\rho(\cdot) : [0,1]^K \to \mathbb{R}$ *that is permutation invariant to the last* $K-1$ *entries, such that* $[\psi(\boldsymbol{Z})]_i = \rho(\boldsymbol{z}_i, \underbrace{\boldsymbol{z}_{i+1}, \cdots, \boldsymbol{z}_K, \cdots, \boldsymbol{z}_{i-1}}_{K-1})$ *for any* $i$.

*Proof.* (Sufficiency) Define $\pi : [K] \to [K]$ be an index mapping associated with the permutation matrix $\boldsymbol{P} \in \Pi(K)$ such that $\boldsymbol{PZ} = \begin{bmatrix} \boldsymbol{z}_{\pi(1)}, \cdots, \boldsymbol{z}_{\pi(K)} \end{bmatrix}^\top$. Then $[\psi(\boldsymbol{z}_{\pi(1)}, \cdots, \boldsymbol{z}_{\pi(K)})]_i = \rho(\boldsymbol{z}_{\pi(i)}, \boldsymbol{z}_{\pi(i+1)}, \cdots, \boldsymbol{z}_{\pi(K)}, \cdots, \boldsymbol{z}_{\pi(i-1)})$. Since $\rho(\cdot)$ is invariant to the last $K-1$ entries, $[\psi(\boldsymbol{PZ})]_i = \rho(\boldsymbol{z}_{\pi(i)}, \boldsymbol{z}_{\pi(i)+1}, \cdots, \boldsymbol{z}_K, \cdots, \boldsymbol{z}_{\pi(i)-1}) = [\psi(\boldsymbol{Z})]_{\pi(i)}$.

(Necessity) Given a permutation-equivariant function $\psi : [0,1]^K \to \mathbb{R}^K$, we first expand it to the following form: $[\psi(\boldsymbol{Z})]_i = \rho_i(\boldsymbol{z}_1, \cdots, \boldsymbol{z}_K)$. Permutation-equivariance means $\rho_{\pi(i)}(\boldsymbol{z}_1, \cdots, \boldsymbol{z}_K) = \rho_i(\boldsymbol{z}_{\pi(1)}, \cdots, \boldsymbol{z}_{\pi(K)})$. Suppose given an index $i$, consider any permutation $\pi : [K] \to [K]$, where $\pi(i) = i$. Then, we have $\rho_i(\boldsymbol{z}_1, \cdots, \boldsymbol{z}_i, \cdots, \boldsymbol{z}_K) = \rho_{\pi(i)}(\boldsymbol{z}_1, \cdots, \boldsymbol{z}_i, \cdots, \boldsymbol{z}_K) = \rho_i(\boldsymbol{z}_{\pi(1)}, \cdots, \boldsymbol{z}_i, \cdots, \boldsymbol{z}_{\pi(K)})$, which implies $\rho_i : \mathbb{R}^K \to \mathbb{R}$ must be invariant to the $K-1$ elements other than the $i$-th element. Now, consider a permutation $\pi$ where $\pi(1) = i$. Then $\rho_i(\boldsymbol{z}_1, \boldsymbol{z}_2, \cdots, \boldsymbol{z}_K) = \rho_{\pi(1)}(\boldsymbol{z}_1, \boldsymbol{z}_2, \cdots, \boldsymbol{z}_K) = \rho_1(\boldsymbol{z}_{\pi(1)}, \boldsymbol{z}_{\pi(2)}, \cdots, \boldsymbol{z}_{\pi(K)}) = \rho_1(\boldsymbol{z}_i, \boldsymbol{z}_{i+1}, \cdots, \boldsymbol{z}_K, \cdots, \boldsymbol{z}_{i-1})$, where the last equality is due to our previous argument. This implies two results. First, for all $i$, $\rho_i(\boldsymbol{z}_1, \boldsymbol{z}_2, \cdots, \boldsymbol{z}_i, \cdots, \boldsymbol{z}_K), \forall i \in [K]$ should be written in terms of $\rho_1(\boldsymbol{z}_i, \boldsymbol{z}_{i+1}, \cdots, \boldsymbol{z}_K, \cdots, \boldsymbol{z}_{i-1})$. Moreover, $\rho_1$ is permutation invariant to its last $K-1$ entries. Therefore, we just need to set $\rho = \rho_1$ and broadcast it accordingly to all entries. We conclude the proof. $\square$

To proceed the proof, we bring in the following mathematical tools (Zaheer et al., 2017):

**Definition 5.** *Given a vector* $\boldsymbol{z} = [\boldsymbol{z}_1, \cdots, \boldsymbol{z}_K]^\top \in \mathbb{R}^K$, *we define power mapping* $\phi_M : \mathbb{R} \to \mathbb{R}^M$ *as* $\phi_M(z) = \begin{bmatrix} z & z^2 & \cdots & z^M \end{bmatrix}^\top$, *and sum-of-power mapping* $\Phi_M : \mathbb{R}^K \to \mathbb{R}^M$ *as* $\Phi_M(\boldsymbol{z}) = \sum_{i=1}^K \phi_M(\boldsymbol{z}_i)$, *where* $M$ *is the largest degree.*

**Lemma 3.** *Let* $\mathcal{X} = \{[\boldsymbol{z}_1, \cdots, \boldsymbol{z}_K]^\top \in [0,1]^K$ *such that* $\boldsymbol{z}_1 < \boldsymbol{z}_2 < \cdots < \boldsymbol{z}_K\}$. *We define mapping* $\tilde{\phi} : \mathbb{R} \to \mathbb{R}^{K+1}$ *as* $\tilde{\phi}(z) = \begin{bmatrix} z^0 & z^1 & z^2 & \cdots & z^K \end{bmatrix}^\top$, *and mapping* $\tilde{\Phi} : \mathbb{R}^K \to \mathbb{R}^{K+1}$ *as* $\tilde{\Phi}(\boldsymbol{z}) = \sum_{i=1}^K \tilde{\phi}(\boldsymbol{z}_i)$, *where* $M$ *is the largest degree. Then* $\tilde{\Phi}$ *restricted on* $\mathcal{X}$, *i.e.,* $\tilde{\Phi} : \mathcal{X} \to \mathbb{R}^{K+1}$, *is a homeomorphism.*

*Proof.* Proved in Lemma 6 in (Zaheer et al., 2017). $\square$

We note that Definition 5 is slightly different from the mappings defined in Lemma 3 (Zaheer et al., 2017) as it removes the constant (zero-order) term. Combining with Lemma 3 (Zaheer et al., 2017) and results in (Wagstaff et al., 2019), we have the following result:

**Lemma 4.** *Let $\mathcal{X} = \{[z_1, \cdots, z_K]^\top \in [0,1]^K \text{ such that } z_1 < z_2 < \cdots < z_K\}$, then there exists a homeomorphism $\Phi_M : \mathcal{X} \to \mathbb{R}^M$ such that $\Phi_M(z) = \sum_{i=1}^K \phi_M(z_i)$ where $\phi_M : \mathbb{R} \to \mathbb{R}^M$ if $M \geq K$.*

*Proof.* For $M = K$, we choose $\phi_K$ and $\Phi_K$ to be the power mapping and power sum with largest degree $K$ defined in Definition 5. We note that $\tilde{\Phi}(z) = \begin{bmatrix} K & \Phi_K(z)^\top \end{bmatrix}^\top$. Since $K$ is a constant, there exists a homeomorphism between the images of $\tilde{\Phi}(z)$ and $\Phi_K(z)$. By Lemma 3, $\tilde{\Phi} : \mathcal{X} \to \mathbb{R}^{K+1}$ is a homeomorphism, which implies $\Phi_K(z) : \mathcal{X} \to \mathbb{R}^K$ is also a homeomorphism.

For $M > K$, we first pad every input $z \in \mathcal{X}$ with a constant $k > 1$ to be an $M$-dimension $\hat{z} \in \mathbb{R}^M$. Note that padding is homeomorphic since $k$ is a constant. All such $\hat{z}$ form a subset $\mathcal{X}' \subset \{[z_1, \cdots, z_M]^\top \in [0,k]^M \text{ such that } z_1 < z_2 < \cdots < z_M\}$. We choose $\hat{\phi} : \mathbb{R} \to \mathbb{R}^M$ to be power mapping and $\hat{\Phi} : \mathcal{X}' \to \mathbb{R}^M$ to be sum-of-power mapping restricted on $\mathcal{X}'$, respectively. Following (Wagstaff et al., 2019), we construct $\Phi_M(z)$ as below:

$$\hat{\Phi}_M(\hat{z}) = \sum_{i=1}^K \hat{\phi}_M(z_i) + \sum_{i=K+1}^M \hat{\phi}_M(k) = \sum_{i=1}^K \hat{\phi}_M(z_i) + \sum_{i=1}^M \hat{\phi}_M(k) - \sum_{i=1}^K \hat{\phi}_M(k) \tag{14}$$

$$= \sum_{i=1}^K (\hat{\phi}_M(z_i) - \hat{\phi}_M(k)) + \sum_{i=1}^M \hat{\phi}_M(k) = \sum_{i=1}^K (\hat{\phi}_M(z_i) - \hat{\phi}_M(k)) + M\hat{\phi}_M(k), \tag{15}$$

which induces $\sum_{i=1}^K (\hat{\phi}_M(z_i) - \hat{\phi}_M(k)) = \hat{\Phi}_M(\hat{z}) - M\hat{\phi}_M(k)$. Let $\phi_M(z) = \hat{\phi}_M(z_i) - \hat{\phi}_M(k)$, and $\Phi_M(z) = \sum_{i=1}^K \phi_M(z_i)$. Since $[0,k]$ is naturally homeomorphic to $[0,1]$, by our argument for $M = K$, $\hat{\Phi}_M : \mathcal{X}' \to \mathbb{R}^M$ is a homeomorphism. This implies $\Phi_M : \mathcal{X} \to \mathbb{R}^M$ is also a homeomorphism. $\qquad\square$

It is straightforward to show the sufficiency of Theorem 1 by verifying that for arbitrary permutation $\pi : [K] \to [K]$, $[\psi(z_{\pi(1)}, \cdots, z_{\pi(K)})]_i = \rho(z_{\pi(i)}, \sum_{j=1}^K \phi(z_j)) = [\psi(z_1, \cdots, z_K)]_{\pi(i)}$. With Lemma 2 and 4, we can conclude the necessity of Theorem 1 by the following construction:

1. By Lemma 2, any permutation equivariant function $\psi(z_1, \cdots, z_K)$ can be written as $[\psi(\cdot)]_i = \tau(z_i, z_{i+1}, \cdots, z_K, \cdots, z_{i-1})$ such that $\tau(\cdot)$ is invariant to the last $K - 1$ elements.

2. By Lemma 4, we know that there exists a homeomorphism mapping $\Phi_K$ which is continuous, invertible, and invertibly continuous. For arbitrary $z \in [0,1]^K$, the difference between $z$ and $\Phi_K^{-1} \circ \Phi_K(z)$ is up to a permutation.

3. Since $\tau(\cdot)$ is permutation invariant to the last $K - 1$ elements, we can construct the function $\tau(z_i, z_{i+1}, \cdots, z_K, \cdots, z_{i-1}) = \tau(z_i, \Phi_{K-1}^{-1} \circ \Phi_{K-1}(z_{i+1}, \cdots, z_K, \cdots, z_{i-1})) = \tilde{\rho}(z_i, \sum_{j \neq i} \phi_{K-1}(z_j)) = \rho(z_i, \sum_{j=1}^K \phi_{K-1}(z_j))$, where $\tilde{\rho}(x, y) = \tau(x, \Phi_{K-1}^{-1}(y))$ and $\rho(x, y) = \tilde{\rho}(x, y - \phi_{K-1}(x))$.

Since $\tau$, $\Phi_{K-1}^{-1}$, and $\phi_{K-1}$ are all continuous functions, their composition $\rho$ is also continuous. $\qquad\square$

**Remark 1.** *Sannai et al. (2019)[Corollary 3.2] showed similar results to our Theorem 1. However, their provided exact form of equivariant set function is written as $\rho(z_i, \sum_{j \neq i} \phi(z_j))$. We note the difference: the summation in Theorem 1 pools over all the nodes inside a hyperedge while the summation in (Sannai et al., 2019)[Corollary 3.2] pools over elements **other than the central node**. To implement the latter equation in hypergraphs which have coupled sets, one has to maintain each node-hyperedge pair, which is computationally prohibitive in practice. An easier proof can be shown by combining results in (Sannai et al., 2019) with our proof technique introduced in the Step 3.*

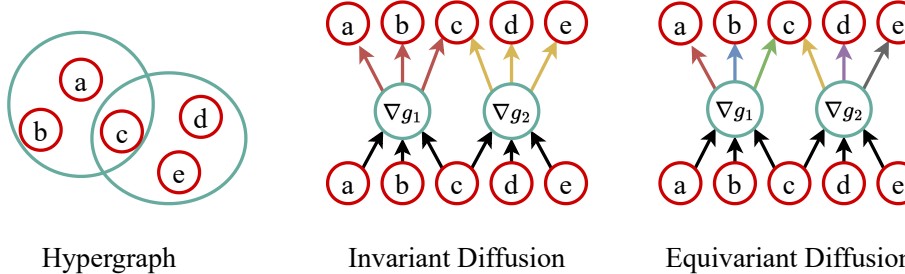

Figure 4: Comparing Invariant Diffusion and Equivariant Diffusion. The direction of the arrows indicates the information flow. The same color of arrows denotes the same passed message while different color of arrows means different passed messages.

## C  ALIGNING ALGORITHM 1 WITH GD/ADMM UPDATES

ED-HNN can be regarded as an unfolding algorithm to optimize objective Eq. 1, where $f$ and $g$ are node and edge potentials, respectively. Essentially, each iteration of ED-HNN simulates a one-step equivariant hypergraph diffusion. Each iteration of ED-HNN has four steps. In the first step, ED-HNN point-wisely transforms the node features via an MLP $\phi$. Next ED-HNN sum-pools the node features onto the associated hyperedges as $\boldsymbol{m}_e = \sum_{v \in e} \phi(\boldsymbol{h}_v)$. The third step is crucial to achieve equivariance, where each node sum-pools the hyperedge features after interacting its feature with every connected hyperedge: $\sum_{e:v \in e} \rho(\boldsymbol{h}_v, \boldsymbol{m}_e)$. The last step is to update the feature information via another node-wse transformation $\varphi$. According to our Theorem 1, $\rho(\boldsymbol{h}_v, \sum_{u \in e} \phi(\boldsymbol{h}_u))$ can represent any equivariant function. Therefore, the messages aggregated from hyperedges $\rho(\boldsymbol{h}_v, \boldsymbol{m}_e)$ can be learned to approximate the gradient $\nabla g$ in Eq. 2.

ED-HNN can well approximate the GD algorithm (Eq. 2), while ED-HNN does not perfectly match ADMM unless we assume $\boldsymbol{Q}_e^{(t)} = \boldsymbol{H}_e^{(t)}$ in Eq. 3 for a practical consideration. This assumption may reduce the performance of the hypergraph diffusion algorithm while our model ED-HNN has already achieved good enough empirical performance. Tracking $\boldsymbol{Q}_e^{(t)}$ means recording the messages from $\mathcal{E}$ to $\mathcal{V}$ for every iteration, which is not supported by the current GNN platforms (Fey & Lenssen, 2019; Wang et al., 2019) and may consume more memory. We leave the study of the algorithms that can track the update of $\boldsymbol{Q}_e^{(t)}$ as a future study. A co-design of the algorithm and the system may be needed to guarantee the scalability of the algorithm.

## D  INVARIANT DIFFUSION VERSUS EQUIVARIANT DIFFUSION

Although our Proposition 2 states hypergraph diffusion operator should be inherently equivariant, many exising frameworks design it to be invariant. In invariant diffusion, hypergraph diffusion operator $\nabla g(\cdot)$ or $\mathbf{prox}\,\eta g(\cdot)$ corresponds to a permutation invariant function rather than a permutation equivariant function (see Definition 4). Mathematically, $\nabla g(\boldsymbol{H}_e) = \mathbf{1}^\top \hat{g}(\boldsymbol{H}_e)$ for some invariant function $\hat{g} : \mathbb{R}^{|e| \times F} \to \mathbb{R}^F$. In contrast to equivariant diffusion, the invariant diffusion operator summarizes the node information to one feature vector and passes this identical message to all nodes uniformly. We provide Fig. 4 to illustrate the key difference between invariant diffusion and equivariant diffusion. Considering implementing such an invariant diffusion by DeepSet (Zaheer et al., 2017), an invariant diffusion should have the following message passing: The difference in

---

**Example: Invariant Diffusion by DeepSet**

**Initialization:** $\boldsymbol{H}^{(0)} = \boldsymbol{X}$.

**For** $t = 0, 1, 2, ..., L-1$, **do:**

1. Designing the messages from $\mathcal{V}$ to $\mathcal{E}$:   $\boldsymbol{m}_{u \to e}^{(t)} = \hat{\phi}(\boldsymbol{h}_u^{(t)})$, for all $u \in \mathcal{V}$.

2. Sum $\mathcal{V} \to \mathcal{E}$ messages over hyperedges $\boldsymbol{m}_e^{(t)} = \sum_{u \in e} \boldsymbol{m}_{u \to e}^{(t)}$, for all $e \in \mathcal{E}$.

3. Broadcast $\boldsymbol{m}_e^{(t)}$ and design the messages from $\mathcal{E}$ to $\mathcal{V}$: $\boldsymbol{m}_{e \to v}^{(t)} = \hat{\rho}(\boldsymbol{m}_e^{(t)})$, for all $v \in e$.

4. Update $\boldsymbol{h}_v^{(t+1)} = \hat{\varphi}(\boldsymbol{h}_v^{(t)}, \sum_{e:u \in e} \boldsymbol{m}_{e \to u}^{(t)}, \boldsymbol{x}_v, d_v)$, for all $v \in \mathcal{V}$.

the third step is worth noting. $\hat{\rho}$ is independent of the target node features such that each hyperedge passes every node an identical message. HyperGCN (Yadati et al., 2019), HGNN (Feng et al., 2019) and other GCNs running on clique expansion can all be reduced to invariant diffusion.

# E    FURTHER DISCUSSION ON OTHER RELATED WORKS

Our ED-HNN is actually inspired by a line of works on optimization-inspired NNs. Optimization-inspired NNs often get praised for their interpretability (Monga et al., 2021; Chen et al., 2021b) and certified convergence under certain conditions of the learned operators (Ryu et al., 2019; Teodoro et al., 2017; Chan et al., 2016). Optimization-inspired NNs mainly focus on the applications such as compressive sensing (Gregor & LeCun, 2010; Xin et al., 2016; Liu & Chen, 2019; Chen et al., 2018), speech processing (Hershey et al., 2014; Wang et al., 2018), image denoising (Zhang et al., 2017b; Meinhardt et al., 2017; Chang et al., 2017; Chen & Pock, 2016), partitioning (Zheng et al., 2015; Liu et al., 2017), deblurring (Schuler et al., 2015; Li et al., 2020b) and so on. Only a few works recently have applied optimization-inspired NNs to graphs as discussed in Sec. 3.4, while our work is the first one to study optimization-inspired NNs on hypergraphs.

Parallel to our hypergraph diffusion inspired HNNs, there are also some models to represent diffusion on graphs. However, as there are only pairwise relations in graphs, the model design is often much simpler than that for hypergraphs. In particular, to construct an expressive operator with for pairwise relations is trivial (corresponding to the case $K = 2$ in Theorem 1). These models unroll either learnable (Yang et al., 2021a; Chen et al., 2021a), fixed $\ell_1$-norm (Liu et al., 2021c) or fixed $\ell_2$-norm pair-wise potentials (Klicpera et al., 2019) to formulate GNNs. Some works also view the graph diffusion as an ODE (Chamberlain et al., 2021; Thorpe et al., 2022), which essentially corresponds to the gradient descent formulation Eq. 2. Implicit GNNs (Dai et al., 2018; Liu et al., 2021a; Gu et al., 2020; Yang et al., 2021b) are to directly parameterize the optimum of an optimization problem while implicit methods are missing for hypergraphs, which is a promising future direction.

# F    ADDITIONAL EXPERIMENTS AND IMPLEMENTATION DETAILS

## F.1    EXPERIMENTS ON ED-HNNII

As we described in Sec. 3.2, an extension to our ED-HNN is to consider $\mathcal{V} \to \mathcal{E}$ message passing and $\mathcal{E} \to \mathcal{V}$ message passing as two equivariant set functions. The detailed algorithm is illustrated in **Algorithm 2**, where the red box highlights the major difference with ED-HNN.

---

**Algorithm 2: ED-HNNII**

**Initialization:** $\boldsymbol{H}^{(0)} = \boldsymbol{X}$ and **three MLPs** $\hat{\phi}$, $\hat{\rho}$, $\hat{\varphi}$ (**shared across $L$ layers**).
**For** $t = 0, 1, 2, ..., L-1$, **do:**

1. Designing the messages from $\mathcal{V}$ to $\mathcal{E}$:    $\boldsymbol{m}_{u \to e}^{(t)} = \hat{\phi}(\boldsymbol{m}_{u \to e}^{(t-1)}, \boldsymbol{h}_u^{(t)})$, for all $u \in \mathcal{V}$.

2. Sum $\mathcal{V} \to \mathcal{E}$ messages over hyperedges $\boldsymbol{m}_e^{(t)} = \sum_{u \in e} \boldsymbol{m}_{u \to e}^{(t)}$, for all $e \in \mathcal{E}$.

3. Broadcast $\boldsymbol{m}_e^{(t)}$ and design the messages from $\mathcal{E}$ to $\mathcal{V}$: $\boldsymbol{m}_{e \to v}^{(t)} = \hat{\rho}(\boldsymbol{h}_v^{(t)}, \boldsymbol{m}_e^{(t)})$, for all $v \in e$.

4. Update $\boldsymbol{h}_v^{(t+1)} = \hat{\varphi}(\boldsymbol{h}_v^{(t)}, \sum_{e:u \in e} \boldsymbol{m}_{e \to u}^{(t)}, \boldsymbol{x}_v, d_v)$, for all $v \in \mathcal{V}$.

---

In our tested datasets, hyperedges do not have initial attributes. In our implementation, we assign a common learnable vector for every hyperedge as their first-layer features $\boldsymbol{m}_{u \to e}^{(0)}$. The performance of ED-HNNII and the comparison with other baselines are presented in Table 4. Our finding is that ED-HNNII can outperform ED-HNN on datasets with relatively larger scale or more heterophily. And the average accuracy improvement by ED-HNNII is around 0.5%. We argue that ED-HNNII inherently has more complex computational mechanism, and thus tends to overfit on small datasets (e.g., Cora, Citeseer, etc.). Moreover, superior performance on heterophilic datasets also implies that injecting more equivariance benefits handling heterophilic data. From Table 4, the measured computational efficiency is also comparable to ED-HNN and other baselines.

Table 4: Additional experiments and updated leaderboard with ED-HNNII. Prediction accuracy (%). **Bold font** highlights when ED-HNNII outperforms the original ED-HNN. Other details are kept consistent with Table 2.

|  | Cora | Citeseer | Pubmed | Cora-CA | DBLP-CA | Training Time ($10^{-1}$ s) |
|---|---|---|---|---|---|---|
| HGNN (Huang & Yang, 2021) | 79.39 ± 1.36 | 72.45 ± 1.16 | 86.44 ± 0.44 | 82.64 ± 1.65 | 91.03 ± 0.20 | 0.24 ± 0.51 |
| HCHA (Bai et al., 2021) | 79.14 ± 1.02 | 72.42 ± 1.42 | 86.41 ± 0.36 | 82.55 ± 0.97 | 90.92 ± 0.22 | 0.24 ± 0.01 |
| HNHN (Dong et al., 2020) | 76.36 ± 1.92 | 72.64 ± 1.57 | 86.90 ± 0.30 | 77.19 ± 1.49 | 86.78 ± 0.29 | 0.30 ± 0.56 |
| HyperGCN (Yadati et al., 2019) | 78.45 ± 1.26 | 71.28 ± 0.82 | 82.84 ± 8.67 | 79.48 ± 2.08 | 89.38 ± 0.25 | 0.42 ± 1.51 |
| UniGCNII (Huang & Yang, 2021) | 78.81 ± 1.05 | 73.05 ± 2.21 | 88.25 ± 0.40 | 83.60 ± 1.14 | 91.69 ± 0.19 | 4.36 ± 1.18 |
| HyperND Tudisco et al. (2021a) | 79.20 ± 1.14 | 72.62 ± 1.49 | 86.68 ± 0.43 | 80.62 ± 1.32 | 90.35 ± 0.26 | 0.15 ± 1.32 |
| AllDeepSets (Chien et al., 2022) | 76.88 ± 1.80 | 70.83 ± 1.63 | 88.75 ± 0.33 | 81.97 ± 1.50 | 91.27 ± 0.27 | 1.23 ± 1.09 |
| AllSetTransformer (Chien et al., 2022) | 78.58 ± 1.47 | 73.08 ± 1.20 | 88.72 ± 0.37 | 83.63 ± 1.47 | 91.53 ± 0.23 | 1.64 ± 1.63 |
| ED-HNN | 80.31 ± 1.35 | 73.70 ± 1.38 | 89.03 ± 0.53 | 83.97 ± 1.55 | 91.90 ± 0.19 | 1.71 ± 1.13 |
| ED-HNNII | 78.47 ± 1.62 | 72.65 ± 1.56 | **89.56 ± 0.62** | 82.17 ± 1.68 | **91.93 ± 0.29** | 1.85 ± 1.09 |
|  | Congress | Senate | Walmart | House | Avg. Rank | Inference Time ($10^{-2}$ s) |
| HGNN (Huang & Yang, 2021) | 91.26 ± 1.15 | 48.59 ± 4.52 | 62.00 ± 0.24 | 61.39 ± 2.96 | 6.00 | 1.01 ± 0.04 |
| HCHA (Bai et al., 2021) | 90.43 ± 1.20 | 48.62 ± 4.41 | 62.35 ± 0.26 | 61.36 ± 2.53 | 6.67 | 1.54 ± 0.18 |
| HNHN (Dong et al., 2020) | 53.35 ± 1.45 | 50.93 ± 6.33 | 47.18 ± 0.35 | 67.80 ± 2.59 | 7.67 | 8.11 ± 0.05 |
| HyperGCN (Yadati et al., 2019) | 55.12 ± 1.96 | 42.45 ± 3.67 | 44.74 ± 2.81 | 48.32 ± 2.93 | 9.22 | 0.87 ± 0.06 |
| UniGCNII (Huang & Yang, 2021) | 94.81 ± 0.81 | 49.30 ± 4.25 | 54.45 ± 0.37 | 67.25 ± 2.57 | 4.56 | 21.22 ± 0.13 |
| HyperND (Tudisco et al., 2021a) | 74.63 ± 3.62 | 52.82 ± 3.20 | 38.10 ± 3.86 | 51.70 ± 3.37 | 6.89 | 0.02 ± 0.01 |
| AllDeepSets (Chien et al., 2022) | 91.80 ± 1.53 | 48.17 ± 5.67 | 64.55 ± 0.33 | 67.82 ± 2.40 | 6.22 | 5.35 ± 0.33 |
| AllSetTransformer (Chien et al., 2022) | 92.16 ± 1.05 | 51.83 ± 5.22 | 65.46 ± 0.25 | 69.33 ± 2.20 | 3.56 | 6.06 ± 0.67 |
| ED-HNN | 95.00 ± 0.99 | 64.79 ± 5.14 | 66.91 ± 0.41 | 72.45 ± 2.28 | 1.56 | 5.87 ± 0.36 |
| ED-HNNII | **95.19 ± 1.34** | 63.81 ± 6.17 | **67.24 ± 0.45** | **73.95 ± 1.97** | 2.67 | 6.07 ± 0.40 |

Table 5: More dataset statistics. CE homophily is the homophily score (Pei et al., 2020) based on CE of hypergraphs.

|  | Cora | Citeseer | Pubmed | Cora-CA | DBLP-CA | Congress | Senate | Walmart | House |
|---|---|---|---|---|---|---|---|---|---|
| # nodes | 2708 | 3312 | 19717 | 2708 | 41302 | 1718 | 282 | 88860 | 1290 |
| # hyperedges | 1579 | 1079 | 7963 | 1072 | 22363 | 83105 | 315 | 69906 | 340 |
| # features | 1433 | 3703 | 500 | 1433 | 1425 | 100 | 100 | 100 | 100 |
| # classes | 7 | 6 | 3 | 7 | 6 | 2 | 2 | 11 | 2 |
| avg. $d_v$ | 1.767 | 1.044 | 1.756 | 1.693 | 2.411 | 427.237 | 19.177 | 5.184 | 9.181 |
| avg. $|e|$ | 3.03 | 3.200 | 4.349 | 4.277 | 4.452 | 8.656 | 17.168 | 6.589 | 34.730 |
| CE Homophily | 0.897 | 0.893 | 0.952 | 0.803 | 0.869 | 0.555 | 0.498 | 0.530 | 0.509 |

## F.2 DETAILS OF BENCHMARKING DATASETS

Our benchmark datasets consist of existing seven datasets (Cora, Citeseer, Pubmed, Cora-CA, DBLP-CA, Walmart, and House) from Chien et al. (2022), and two newly introduced datasets (Congress (Fowler, 2006a) and Senate (Fowler, 2006b)). For existing datasets, we downloaded the processed version by Chien et al. (2022). For co-citation networks (Cora, Citeseer, Pubmed), all documents cited by a document are connected by a hyperedge (Yadati et al., 2019). For co-authorship networks (Cora-CA, DBLP), all documents co-authored by an author are in one hyperedge (Yadati et al., 2019). The node features in citation networks are the bag-of-words representations of the corresponding documents, and node labels are the paper classes. In the House dataset, each node is a member of the US House of Representatives and hyperedges group members of the same committee. Node labels indicate the political party of the representatives. (Chien et al., 2022). In Walmart, nodes represent products purchased at Walmart, hyperedges represent sets of products purchased together, and the node labels are the product categories (Chien et al., 2022). For Congress (Fowler, 2006a) and Senate(Fowler, 2006b), we used the same setting as (Veldt et al., 2022). In Congress dataset, nodes are US Congresspersons and hyperedges are comprised of the sponsor and co-sponsors of legislative bills put forth in both the House of Representatives and the Senate. In Senate dataset, nodes are US Congresspersons and hyperedges are comprised of the sponsor and co-sponsors of bills put forth in the Senate. Each node in both datasets is labeled with political party affiliation. Both datasets were from James Fowler's data (Fowler, 2006b;a). We also list more detailed statistical information on the tested datasets in Table 5.

## F.3 HYPERPARAMETERS FOR BENCHMARKING DATASETS

For a fair comparison, we use the same training recipe for all the models. For baseline models, we precisely follow the hyperparameter settings from (Chien et al., 2022). For ED-HNN, we adopt Adam optimizer with fixed learning rate=0.001 and weight decay=0.0, and train for 500 epochs for all datasets. The standard deviation is reported by repeating experiments on ten different data splits. We fix the input dropout rate to be 0.2, and dropout rate to be 0.3. For internal MLPs, we add a

LayerNorm for each layer similar to (Chien et al., 2022). Other parameters regarding model sizes are obtained by grid search, which are enumerated in Table 6. The search range of layer number is $\{1, 2, 4, 6, 8\}$ and the hidden dimension is $\{96, 128, 256, 512\}$. We find the model size is proportional to the dataset scale, and in general heterophilic data need deeper architecture. For ED-HNNII, due to the inherent model complexity, we need to prune model depth and width to fit each dataset.

---

**Algorithm 3: Contextual Hypergraph Stochastic Block Model**

**Initialization:** Empty hyperedge set $\mathcal{E} = \emptyset$. Draw vertex set $\mathcal{V}_1$ of 2,500 nodes with class 1. Draw vertex set $\mathcal{V}_2$ of 2,500 nodes with class 2.
**For** $i = 0, 1, 2, ..., 1,000,$ **do:**
    1. Sample a subset $e_1$ with $\alpha_1$ nodes from $\mathcal{V}_1$.
    2. Sample a subset $e_2$ with $\alpha_2$ nodes from $\mathcal{V}_2$.
    3. Construct the hyperedge $\mathcal{E} \leftarrow \mathcal{E} \cup \{e_1 \cup e_2\}$.

---

Table 6: Choice of hyperparameters for ED-HNN. # is short for "number of", hd. stands for hidden dimension, cls. means the classifier. When number of MLP layers equals to 0, the MLP boils down to be an identity mapping.

| ED-HNN | Cora | Citeseer | Pubmed | Cora-CA | DBLP-CA | Congress | Senate | Walmart | House |
|---|---|---|---|---|---|---|---|---|---|
| # iterations | 1 | 1 | 8 | 1 | 1 | 4 | 8 | 6 | 8 |
| # layers of $\hat{\phi}$ | 0 | 0 | 2 | 1 | 1 | 2 | 2 | 2 | 2 |
| # layers of $\hat{\rho}$ | 1 | 1 | 2 | 1 | 1 | 2 | 2 | 2 | 2 |
| # layers of $\hat{\varphi}$ | 1 | 1 | 1 | 1 | 1 | 2 | 2 | 2 | 2 |
| # layer cls. | 1 | 1 | 2 | 2 | 2 | 2 | 2 | 2 | 2 |
| MLP hd. | 256 | 256 | 512 | 128 | 128 | 156 | 512 | 512 | 256 |
| cls. hd. | 256 | 256 | 256 | 96 | 96 | 128 | 256 | 256 | 128 |
| ED-HNNII | Cora | Citeseer | Pubmed | Cora-CA | DBLP-CA | Congress | Senate | Walmart | House |
| # iterations | 1 | 1 | 4 | 1 | 1 | 4 | 4 | 4 | 4 |
| # layers of $\hat{\phi}$ | 1 | 1 | 2 | 1 | 1 | 2 | 2 | 2 | 2 |
| # layers of $\hat{\rho}$ | 1 | 1 | 2 | 1 | 1 | 2 | 2 | 2 | 2 |
| # layers of $\hat{\varphi}$ | 0 | 1 | 1 | 1 | 1 | 1 | 1 | 1 | 1 |
| # layer cls. | 1 | 1 | 2 | 2 | 2 | 2 | 2 | 2 | 2 |
| MLP hd. | 128 | 128 | 512 | 128 | 128 | 156 | 512 | 512 | 256 |
| cls. hd. | 96 | 96 | 256 | 96 | 96 | 128 | 256 | 256 | 128 |

## F.4 SYNTHETIC HETEROPHILIC DATASETS

We use the contextual hypergraph stochastic block model (Deshpande et al., 2018; Ghoshdastidar & Dukkipati, 2014; Lin & Wang, 2018) to synthesize data with controlled heterophily. The generated graph contains 5,000 nodes and two classes in total, and 2,500 nodes for each class. We construct hyperedges by randomly sampling $\alpha_1$ nodes from class 1, and $\alpha_2$ nodes from class 2 without replacement. Each hyperedge has a fixed cardinality $|e| = \alpha_1 + \alpha_2 = 15$. We draw 1,000 hyperedges in total. The detailed data synthesis pipeline is summarized in **Algorithm 3**. We use $\alpha = \min\{\alpha_1, \alpha_2\}$ to characterize the heterophily level of the hypergraph. For a more intuitive illustration, we list the CE homophily corresponding to different $\alpha$ in Table 7. Experiments on synthetic

Table 7: Correspondence between Heterophily $\alpha$ and CE Homophily.

| $\alpha$ | 1 | 2 | 3 | 4 | 6 | 7 |
|---|---|---|---|---|---|---|
| CE Homophily | 0.875 | 0.765 | 0.672 | 0.596 | 0.495 | 0.474 |

heterophilic datasets fix the training hyperparameters and hidden dimension=256 to guarantee a fair parameter budget (~1M). Baseline HNNs are all of one-layer architecture as they are not scalable with the depth shown in Sec. 4.3. Since our ED-HNN adopts parameter sharing scheme, we can easily repeat the diffusion layer twice to achieve better results without parameter overheads.

## F.5 SYNTHETIC DIFFUSION DATASETS AND ADDITIONAL EXPERIMENTS

In order to evaluate the ability of ED-HNN to express given hypergraph diffusion, we generate semi-synthetic diffusion data using the Senate hypergraph (Chodrow et al., 2021) and synthetic node

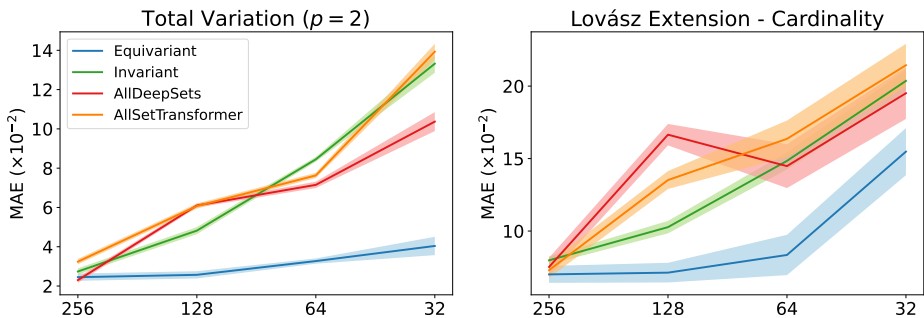

Figure 5: Comparing the Powers to Represent known Diffusion (using ADMM with the proximal operators in Eq. 3 and 4): MAE v.s. Latent Dimensions

features. The data consists of 1,000 pairs $(\boldsymbol{H}^{(0)}, \boldsymbol{H}^{(1)})$. The initial node features $\boldsymbol{H}^{(0)}$ are sampled from 1-dim Gaussian distributions with mean 0 and variance randomly drawn between 1 and 100. That is, to generate a single instance of $\boldsymbol{H}^{(0)}$, we first pick $\sigma$ uniformly from [1,10], and then sample the coordinate entries as $\boldsymbol{h}_v^{(0)} \sim N(0, \sigma^2)$. Then we apply the gradient step in Eq. 2 to obtain the corresponding $\boldsymbol{H}^{(1)}$. For non-differentiable node or edge potentials, we adopt subgradients for convenient computation. We fix the node potential as $f(\boldsymbol{h}_v; \boldsymbol{x}_v) = (\boldsymbol{h}_v - \boldsymbol{x}_v)^2$ where $\boldsymbol{x}_v \equiv \boldsymbol{h}_v^{(0)}$. We consider 3 different edge potentials from Example 1 with varying complexities: a) CE, b) TV ($p = 2$) and c) LEC ($p = 2$). For LEC, we set $\boldsymbol{y}$ as follows: if $|e|$ is even, then $y_i = 2/|e|$ if $i \le |e|/2$ and $-2/|e|$ otherwise; if $|e|$ is odd, then $y_i = 2/(|e|-1)$ if $i \le (|e|-1)/2$, $y_i = 0$ if $i = (|e|+1)/2$, and $y_0 = -2/(|e|-1)$ otherwise. In order to apply the gradient step in Eq. 2 we need to specify the learning rate $\eta$. We choose $\eta$ in a way such that $\mathrm{Var}(\boldsymbol{H}^{(1)})/\mathrm{Var}(\boldsymbol{H}^{(0)})$ does not vary too much among the three different edge potentials. Specifically, we set $\eta = 0.5$ for CE, $\eta = 0.02$ for TV and $\eta = 0.1$ for LEC.

Beyond semi-synthetic diffusion data generated from the gradient step Eq. 2, we also considered synthetic diffusion data obtained from the proximal operators Eq. 3 and 4. We generated a random uniform hypergraph with 1,000 nodes and 1,000 hyperedges of constant hyperedge size 20. The diffusion data on this hypergraph consists of 1,000 pairs $(\boldsymbol{H}^{(0)}, \boldsymbol{H}^{(1)})$. The initial node features $\boldsymbol{H}^{(0)}$ are sampled in the same way as before. We apply the updates given by Eq. 3 and 4 to obtain $\boldsymbol{H}^{(1)}$. We consider the same node potential and 2 edge potentials TV ($p = 2$) and LEC ($p = 2$). We set $\eta = 1/2$ for both cases. We show the results in Figure 5. The additional results resonate with our previous results in Figure 3. Again, our ED-HNN outperforms other baseline HNNs by a significant margin when hidden dimension is limited.

### F.6 MORE COMPLEXITY ANALYSIS

In this section, we provide more efficiency comparison with HyperSAGE (Arya et al., 2020) and LEGCN (Yang et al., 2020). As we discussed in 3.4, HyperSAGE and LEGCN may be able to learn hyperedge equivariant operator. However, both HyperSAGE and LEGCN need to build node-hyperedge pairs, which is memory consuming and unfriendly to efficient message passing implementation. To demonstrate this point, we compare our model with HyperSAGE and LEGCN in terms of training and inference efficiency. We reuse the official code provided in OpenReview to benchmark HyperSAGE. Since LEGCN has not released code, we reimplement it using Pytorch Geometric Library. We can only test the speed of these models on Cora dataset, because HyperSAGE and LEGCN cannot scale up as there is no efficient implementation yet for their computational prohibitive preprocessing procedures. The results are presented in Table 8. We note that HyperSAGE cannot easily employ message passing between $\mathcal{V}$ and $\mathcal{E}$, since neighbor aggregation step in Hyper-SAGE needs to rule out the central node. Its official implementation adopts a naive "for"-loop for each forward pass which is unable to fully utilize the GPU parallelism and significant deteriorates the speed. LEGCN can be implemented via message passing, however, the graphs expanded over node-edge pairs are much denser thus cannot scale up to larger dataset.

Table 8: Performance and Efficiency Comparison with HyperSAGE and LEGCN. The prediction accuracy of HyperSAGE is copied from the original manuscript (Arya et al., 2020).

|  | HyperSAGE (Arya et al., 2020) | LEGCN (Yang et al., 2020) | ED-HNN | ED-HNNII |
|---|---|---|---|---|
| Training Time ($10^{-1}$ s) | $43.93 \pm 2.15$ | $0.56 \pm 0.71$ | $0.15 \pm 0.68$ | $0.25 \pm 0.48$ |
| Inference Time ($10^{-2}$ s) | $297.57 \pm 30.57$ | $0.42 \pm 0.06$ | $0.15 \pm 0.03$ | $0.20 \pm 0.08$ |
| Prediction Accuracy (%) | $69.30 \pm 2.70$ | $73.34 \pm 1.06$ | $80.31 \pm 1.35$ | $78.47 \pm 1.62$ |

Table 9: Sensitivity Test on Hidden Dimension on Pubmed Dataset.

| Model | 512 | 256 | 128 | 64 |
|---|---|---|---|---|
| AllDeepSets (Chien et al., 2022) | $88.75 \pm 0.33$ | $88.41 \pm 0.37$ | $87.50 \pm 0.42$ | $86.78 \pm 0.40$ |
| AllSetTransformer (Chien et al., 2022) | $88.72 \pm 0.37$ | $88.16 \pm 0.24$ | $87.36 \pm 0.23$ | $86.21 \pm 0.25$ |
| ED-HNN (ours) | $89.03 \pm 0.53$ | $88.74 \pm 0.38$ | $88.84 \pm 0.38$ | $88.76 \pm 0.24$ |

## F.7 SENSITIVITY TO HIDDEN DIMENSIONS

We conduct expressivity vs hidden size experiments in Tab. 9, where we control the hidden dimension of hypergraph models and test their performance on Pubmed dataset. We choose top-performers AllDeepSets and AllSetTransformer as the compared baselines. On real-world data, our 64-width ED-HNN can even achieve on-par results with 512-width AllSet models. This implies our ED-HNN has better tolerance of low hidden dimension and we owe this to higher expressive power by achieving an equivariance.

