# OpenReview forum: "Equivariant Hypergraph Diffusion Neural Operators"
_ICLR.cc/2023/Conference — ICLR 2023 poster_

### Official Review · Reviewer_GNhA · 2022-10-16

**Confidence:** 5
**Correctness:** 3
**Technical Novelty And Significance:** 3
**Empirical Novelty And Significance:** 2
**Recommendation:** 6

**Clarity, Quality, Novelty And Reproducibility:**

\
**Clarity**

While the paper is well-organised, the clarity of the paper can be improved.

Specifically, the authors claim that Step 3 in Algorithm 1 is a simple yet significant contribution for the hypergraph neural network community, however, it does seem marginal when viewed from a graph perspective.

In other words, the star expansion of a hypergraph can be viewed as a bipartite graph and from this perspective, Step 3 in Algorithm 1 is marginal and is well-known in the message passing community.

\
**Quality**

Most (if not all) real-world hypergraphs considered in the paper are those which contain a significant fraction of higher-order hyperedges (i.e., size >= 3).

The quality of the paper can be improved by discussing hypergraphs in which a vast majority of hyperedges are of size 1 or 2 and which contain an insignificant fraction of higher-order hyperedges, e.g., contact networks, online forum thread networks [1].

The concern here is that graph-based methods (on clique expansion) would be strong baselines on such datasets and it is unclear if equivariance will play a significant role.

[1] How Do Hyperedges Overlap in Real-World Hypergraphs? -- Patterns, Measures, and Generators, In WebConf'21



\
**Novelty**

The theory connecting equivariance, hypergraph diffusion, and hypergraph neural networks is somewhat new.

Having said that, a relevant (concurrent) work, that needs to be positioned and differentiated with, is the following:

Equivariant Hypergraph Neural Networks, In ECCV'22.


\
**Reproducibility**

The main part and the supplementary part include enough material, e.g., dataset details, proofs, baselines with references, hyperparameters, for an expert to replicate the results of the paper.

___

**Strength And Weaknesses:**

\
**Strengths**

\+ The paper is well-organised and relevant to the ICLR community.

\+ The theory in the paper significantly establishes connections between HNNs and hypergraph diffusion.

\+ ED-HNN works well in practice on several node classification datasets in comparison with several existing HNNs.


\
**Weaknesses**

\- The key innovation in the design of ED-HNN (i.e., Step 3 in Algorithm 1) is marginal with respect to prior art.

\- ED-HNN (and message passing on star expansion) can potentially be problematic for real-world hypergraphs in which the vast majority of hyperedges are of size two, e.g., contact networks, online forum thread networks.

___

**Summary Of The Paper:**

The paper proposes Equivariant Diffusion-Hypergraph Neural Network (ED-HNN) for hypergraph structured data.

ED-HNN is implemented as a message-passing neural network on the star expansion of the hypergraph and can provably represent any continuous hypergraph diffusion operator.

ED-HNN competes with recent HNNs on hypergraph node classification datasets.

___

**Summary Of The Review:**

While the proposed method is theoretically grounded and effective on standard benchmarks, the paper can be improved in terms of clarity and quality.

___

---

> ### Author Response · Authors · 2022-11-14
> **Response to Reviewer GNhA**
>
> Dear Reviewer GNhA,
>
> We thank reviewer GNhA for acknowledge our work’s significance in HNN community and giving us a favorable initial evaluation. Per Reviewer GNhA’s questions on our architectural design and experiments, please see our responses are as follows:
>
> **1. The key innovation in the design of ED-HNN (i.e., Step 3 in Algorithm 1) is marginal from the perspective of the simple graph.**
>
> We respectfully argue that this statement may be negotiable. We are aware that the step 3 in Algorithm 1 is simply to apply a message passing neural network [1] to the bipartite graphs induced by hypergraphs. However, “the change is simple” does not mean “the contribution is marginal”. The key technical contribution of this work is to show that even with such a simple  architecture, one may be able to well approximate a big class of hypergraph diffusions via scalable neural network implementation. To the best of our knowledge, we are to first one to make such connections, which substantially reduces the required architecture complexity in [2,3,4] to represent some specific hypergraph diffusions. Our experiments in Tab 2 also verify the effectiveness of this simple architecture in real-world datasets.
>
>
>
> [1] Gilmer et al. Neural message passing for quantum chemistry, ICML 2017.
> [2] Arya et al. Hypersage: Generalizing inductive representation learning on hypergraphs.
> [3] Yadati et al. Neural message passing for multi-relational ordered and recursive hypergraphs. NeurIPS, 2020
> [4] Y. Yang et al. Graph neural networks inspired by classical iterative algorithms. ICML 2021.
>
> **2. ED-HNN can potentially be problematic for real-world hypergraphs in which the vast majority of hyperedges are of size two.**
>
> Our method is proposed to be a universal backbone which should fit any data in principle. We indeed conducted experiments on those hypergraphs with sparse edges. As we show in Table 1, citation networks in Cora, Citeseer, Pubmed, Cora-CA datasets only contain 1 - 2 nodes on average. Our performance presented in Table 2 demonstrates that our ED-HNN can effectively deal with such hypergraphs and reach superior results on ALL these datasets.
>
> To be more convincing, we also test our ED-HNN on some simple graph datasets (i.e., the size of each edge is 2) and the results are present below:
>
> | Method | Cora | Citeseer | Pubmed | Texas | Wisconsin | Cornell | Actor |
> | ------------- | ------------- | ------------- | ------------- | ------------- | ------------- | ------------- | ------------- |
> | GCN | 80.68$\pm$0.31 | 71.68$\pm$0.56 | 79.89$\pm$0.23 | 56.67$\pm$2.45 | 57.84$\pm$0.98 | 53.67$\pm$4.21 | 28.84$\pm$0.12 |
> | GAT | 81.24$\pm$0.65 | 71.03$\pm$0.90 | 77.75$\pm$0.74 | 59.49$\pm$4.21 | 52.35$\pm$2.49 | 49.08$\pm$5.10 | 28.23$\pm$0.35 |
> | GraphSAGE | 81.34$\pm$0.39 | 70.15$\pm$0.65 | 78.21$\pm$0.58 | 73.78±2.11 | 74.31$\pm$1.63 | 70.59$\pm$3.41 | 35.64$\pm$0.22 |
> | HyperGCN | 76.63$\pm$0.15 | 68.82$\pm$2.14 |76.38$\pm$1.51 | 54.95$\pm$5.63 | 51.46$\pm$3.12 | 46.33 ± 9.72 | 30.42$\pm$0.54 |
> | HGNN | 79.60$\pm$0.44 | 70.13$\pm$1.07 | 78.33$\pm$0.58 | 59.46$\pm$4.68 | 62.09$\pm$1.13 | 50.45$\pm$4.13 | 33.09$\pm$0.86 |
> | AllDeepSets | 73.60$\pm$0.53 | 63.93$\pm$1.16 | 68.50$\pm$8.94 | 65.77$\pm$5.63 | 48.37 $\pm$ 9.06 | 49.55$\pm$5.63 | 35.22$\pm$0.83 |
> | AllSetTransformer | 72.73$\pm$1.47 | 65.90$\pm$0.69 | 75.63$\pm$0.65 | 63.06$\pm$5.63 | 52.29$\pm$10.80 | 45.95$\pm$7.15 | 34.67$\pm$0.99 |
> | ED-HNN | 81.92$\pm$0.35 | 72.64$\pm$0.69 | 80.67$\pm$0.54 | 78.11$\pm$1.89 | 75.69$\pm$2.18 | 69.64$\pm$3.22 | 36.92$\pm$0.33 |
>
> The ED-HNN has good performance compared GNNs and other HNNs, especially when handling heterophilic datasets. These results further support the universality of ED-HNN to different types of graphs.
>
>
> **3. Graph-based methods (on clique expansion) would be strong baselines on such datasets and it is unclear if equivariance will play a significant role.**
>
> Actually HGNN compared in our Table 2 is equivalent to a GCN running on the clique expansion. As the results indicate, our ED-HNN largely outperforms graph-based methods on clique expansion. Moreover, we also compare ED-HNN on common graph datasets above, which also achieves good performance.
>
> Best,
>
> Paper #2887 Authors

---

> > ### Comment · Reviewer_GNhA · 2022-11-14
> > **Follow-up**
> >
> > Thanks for the response and the additional experiments. I have read all the reviews and their responses.
> >
> > One weakness of the paper is that it was unclear if ED-HNN was effective on real-world hypergraphs in which the vast majority of hyperedges are of size two, e.g., contact networks, online forum thread networks. In the response, the authors have conducted experiments on standard graph datasets (in which all hyperedges are of size two) instead. I had a few follow-up thoughts regarding the new experiments.
> >
> > 1. I was wondering why the performances of GCN and HGNN were different in the new table (presented in the author response) if "HGNN is equivalent to a GCN running on the clique expansion."
> > 2. I was curious to know if the authors had an explanation/intuition why ED-HNN outperforms traditional GNNs such as GCN on Cora, Citeseer, Pubmed, which, also pointed out by Reviewer TN8f, are standard citation graph datasets.
> > 3. In Table 1 of this submission, the average hyperedge size is less than 2 for the co-citation versions of Cora, Citeseer, Pubmed which means that, on average, a document cites fewer than 2 documents. The sizes seem inconsistent from those reported in prior work, e.g., Table 3 (full dataset statistics) on Page 21 in the AllSet paper [1]. Please clarify if I have missed anything.
> >
> > [1] [Eli Chien et al. You are allset: A multiset function framework for hypergraph neural networks, ICLR 2022](https://openreview.net/forum?id=hpBTIv2uy_E)

---

> > > ### Author Response · Authors · 2022-11-15
> > > **Responses to Reviewer GNhA's Follow-up**
> > >
> > > Dear GNhA,
> > >
> > > Thanks for your prompt reply and very insightful questions. Our clarification on our new experiments are as below:
> > >
> > > **1. Why the performances of GCN and HGNN were different in the new table?**
> > >
> > > Conceptually, HGNN is identical to running GCNs on hypergraph expansion. However, in practice, HGNN are implemented by message passing on hypergraph representation of simple graphs (no self loops), which makes some difference. Moreover, HGNN and GCN have different normalization schemes. Specifically, after clique expansion in HGNN, the resultant adjacency matrix has form $D_v^{-1/2}HD_e^{-1}H^TD_v^{-1/2}$, which is not exactly (though conceptually similar) equivalent to GCN’s $D^{-1/2}(A+I)D^{-1/2}$ as $HD_e^{-1}H^T \ne A+I$. Moreover, GCN typically adopts a renormalization trick, which HGNN does not have in our implementation adopted from AllSet [1].
> > >
> > > [1] Eli Chien et al. You are allset: A multiset function framework for hypergraph neural networks, ICLR 2022.
> > >
> > > **2. Why does ED-HNN outperform traditional GNNs such as GCN on Cora, Citeseer, Pubmed?**
> > >
> > > In our additional experiments (in the previous response), we report performance on Cora, Citeseer, Pubmed dataset of **simple graphs**. ED-HNN outperforms GCN because it introduces new weights to remix central node features and neighborhood features. In simple graph regime, this amounts to the architecture adopted in [1][2], which are already shown to have better performance than vanilla GCN in existing literature.
> > > As we clarified in our response to Reviewer TN8f and our Appendix F.2 in revision, the tested citation networks in our Tab. 2 (in the paper) are **hypergraph datasets** instead of standard simple graph benchmarks. For co-citation networks (Cora, Citeseer, Pubmed), all documents cited by a document are connected by a hyperedge. For co-authorship networks (Cora-CA, DBLP), all documents co-authored by an author are in one hyperedge. In short, ED-HNN outperforms GNNs on clique expansion (e.g., HyperGCN, HGNN) because it can achieve equivariant message passing, which is more expressive than invariant message passing in HyperGCN and HGNN.
> > >
> > > [1] Gilmer et al. Neural message passing for quantum chemistry, ICML 2017.
> > >
> > > [2] Chen et al. Simple and Deep Graph Convolutional Networks, ICML 2020
> > >
> > > **3. The sizes seem inconsistent from those reported in prior work, e.g., Table 3 (full dataset statistics) on Page 21 in the AllSet paper.**
> > >
> > > Thanks reviewer for this nice catch. The inconsistency in dataset statistics is caused by the different calculation methods. First of all, we follow the data preprocessing of AllSet throughout the whole paper for fair comparison. Such data preprocessing always adds a self-loop to each node, which is a hyperedge with only one node inside. In our paper, we compute the statistics on the hypergraph with such self-loops, while in the AllSet paper, they compute the statistics before adding such self-loops. That is why our hyperedge size can seem to be smaller than 2, while the node degrees are 1 larger than AllSet reported numbers. Therefore, before adding self-loop, the average hyperedge size should follow AllSet paper, which is around 3-4. Sorry for the confusion. We have fixed this issue in our revision.
> > >
> > > Even though hyperedge size in Cora, Citeseer, Pubmed is 3-4 on average, we observe that more than 50% hyperedges in those hypergraphs are of size 2. Therefore, our ED-HNN’s advantageous performance on these three datasets can still imply its effectiveness of handling sparse hyperedges.
> > >
> > > Best,
> > >
> > > Paper #2887 Authors

---

> > > > ### Comment · Reviewer_GNhA · 2022-11-16
> > > > **Thanks for the Fixes**
> > > >
> > > > Thanks again for the clarification. The clarity of the paper can still be improved.
> > > >
> > > > Since renormalisation trick (and using $A+I$ instead of $A$) is a well-known idea (and a straightforward component) for GNNs such as GCN/HGNN/HyperGCN, the reasons for benefits of ED-HNN over its baselines should be more clearly explained following all the tables of the submission (including the new table presented in the response).
> > > >
> > > > For example, ED-HNN seems to have at least three features that do not exist in HGNN
> > > > * *equivariance* which is the key contribution of the paper (step 3 in Algorithm 1),
> > > > * *renormalisation trick* (and using $A+I$ instead of $A$) which is a simple well-known idea that can be easily added to baselines,
> > > > *   *new weights* to remix central node features and neighborhood features which is another well-known idea
> > > >
> > > >
> > > > It is still unclear why GCN and HyperGCN results should be different in the new table since HyperGCN includes the renormalisation trick. For graph datasets, HyperGCN and GCN seem exactly the same models.
> > > >
> > > > There seems to be another small change which can dramatically alter the empirical results: *the hypergraph representation of the graph dataset* which, to my understanding, includes self-loops to each node and creates a hyperedge of size one (for each node). To my understanding, both HGNN and HyperGCN, in their original papers, did not create self-loop hyperedges of size one.
> > > >
> > > > For a fairer comparison, ED-HNN needs to be compared against the best models. It should be very clear how much benefit equivariance (the main contribution of the paper) has over simple well-known tricks such as renormalisation trick, new weights  to remix central node features and neighborhood features, and (presence or absence of) hypergraph representation. Alternatively, an ED-HNN ablation study of such features would also give us some insights.

---

> > > > > ### Author Response · Authors · 2022-11-17
> > > > > **Response to Reviewer GNhA**
> > > > >
> > > > > Dear Reviewer GNhA,
> > > > >
> > > > > We greatly appreciate your rigorous attitude towards doing research. We are sorry for the confusion. Here we would like to clarify the implementation details to resolve the confusion.
> > > > > 1. First of all, when doing all the experiments in the paper including Tab.2, we added self-loops to all the baselines. The way to add self-loops on a hypergraph is to construct a hyperedge for each node in the hypergraph, where each of these added hyperedges only contains one node. Every method (including ED-HNN and baselines) uses the same self-loop trick. Therefore, the comparisons are fair in all of our hypergraph experiments from the perspective of having self-loops or not.
> > > > > 2. Nevertheless, in the newly added simple0graph experiments, we did not add self-loops which we did so fast to match the response timeline. We feel sorry for that. We fixed this problem and obtained the following table.
> > > > >
> > > > > | Method | Cora | Citeseer | Pubmed | Texas | Wisconsin | Cornell | Actor |
> > > > > | ------------- | ------------- | ------------- | ------------- | ------------- | ------------- | ------------- | ------------- |
> > > > > | GCN | 80.68$\pm$0.31 | 71.68$\pm$0.56 | 79.89$\pm$0.23 | 56.67$\pm$2.45 | 57.84$\pm$0.98 | 53.67$\pm$4.21 | 28.84$\pm$0.12 |
> > > > > | GAT | 81.24$\pm$0.65 | 71.03$\pm$0.90 | 77.75$\pm$0.74 | 59.49$\pm$4.21 | 52.35$\pm$2.49 | 49.08$\pm$5.10 | 28.23$\pm$0.35 |
> > > > > | GraphSAGE | 81.34$\pm$0.39 | 70.15$\pm$0.65 | 78.21$\pm$0.58 | 73.78$\pm$2.11 | 74.31$\pm$1.63 | 70.59$\pm$3.41 | 35.64$\pm$0.22 |
> > > > > | HyperGCN | 76.63$\pm$0.15 | 68.82$\pm$2.14 |76.38$\pm$1.51 | 54.95$\pm$5.63 | 51.46$\pm$3.12 | 46.33 $\pm$ 9.72 | 30.42$\pm$0.54 |
> > > > > | HyperGCN + Self-loop | 80.43$\pm$0.26 | 71.59$\pm$0.48 | 79.65$\pm$0.61 | 56.74$\pm$2.04 | 57.66$\pm$1.24 | 53.85$\pm$3.98 | 29.07$\pm$0.14 |
> > > > > | HGNN | 79.60$\pm$0.44 | 70.13$\pm$1.07 | 78.33$\pm$0.58 | 59.46$\pm$4.68 | 62.09$\pm$1.13 | 50.45$\pm$4.13 | 33.09$\pm$0.86 |
> > > > > | HGNN + Self-loop | 80.60$\pm$0.35 | 71.67$\pm$0.79 | 79.53$\pm$0.46 | 57.35$\pm$2.38 | 58.48$\pm$0.89 | 52.86$\pm$4.76 | 29.23$\pm$0.66 |
> > > > > | ED-HNN | 81.92$\pm$0.35 | 72.64$\pm$0.69 | 80.67$\pm$0.54 | 78.11$\pm$1.89 | 75.69$\pm$2.18 | 69.64$\pm$3.22 | 36.92$\pm$0.33 |
> > > > >
> > > > > Now the gaps between HGNN+self-loop
> > > > > / HyperGCN+self-loop and GCN are just marginal. Regarding the renormalization trick, HGNN and HyperGCN have their own remormalization tricks. These are the parts of their model design and we do not see a principled way to adjust them. Also, one thing to note is that HyperGCN will reduce to GCN mathematically over simple graphs. HGNN will not reduce to GCN exactly over simple graphs due to different renormalization weights. However, their performances on simple graphs are close.
> > > > >
> > > > > To further verify the importance of equivariance diffusion, we also conduct an ablation study on our equivariance design on all the hypergraph datasets. Specifically, we replace the Step 3 in Algorithm 1 by removing $h_v^{(t)}$ from $\rho$, and keep all remaining parts of the model unchanged. The comparison is presented as below:
> > > > >
> > > > > | Method | Cora | Citeseer | Pubmed | Cora-CA | DBLP-CA | Congress | Senate | Walmart | House |
> > > > > | ------------- | ------------- | ------------- | ------------- | ------------- | ------------- | ------------- | ------------- | ------------- | ------------- |
> > > > > | ED-HNN w/o Equiv. | 77.20 $\pm$ 1.15 | 71.65 $\pm$ 1.49 | 87.73 $\pm$ 0.64 | 81.91 $\pm$ 1.33  | 90.72 $\pm$ 0.27  | 87.52 $\pm$ 1.49 | 48.25 $\pm$ 4.87 | 62.80 $\pm$ 0.63 | 65.40 $\pm$ 2.59 |
> > > > > | ED-HNN  |80.31 $\pm$ 1.35 | 73.70 $\pm$ 1.38 | 89.03 $\pm$ 0.53 | 83.97 $\pm$ 1.55 | 91.90 $\pm$ 0.19 | 95.00 $\pm$ 0.99 | 64.79 $\pm$ 5.14 | 66.91 $\pm$ 0.41 | 72.45 $\pm$ 2.28 |
> > > > >
> > > > > These ablation results support that our key design is prominent. Replacing step 3 in the Algorithm with an invariant diffusion can largely degrade the ED-HNN performance, in particular for the last four heterophilic hypergraphs. Also, we observe some gaps between  ED-HNN w/o Equiv. and other invariant models, say HGNN. We guess this is due to the significant drop in the model expressive power as now we adopt both invariant layers and further ask to share parameters across layers (note this is essential to implement an actual hypergraph diffusion process).
> > > > >
> > > > > The reviewer also mentioned adjusting remix weights. We are not sure how to change remix weights in ED-HNN with the goal of doing a fair comparison. We adopt the most standard way to do mixing: add self-loops (by following AllSet) and adopt mean aggregation over the bipartite representation of hypergraphs. We do not compute any extra weights. If the reviewer has a particular proposal in mind on how to adjust remix weights in principle for a fair comparison, please let us know. We want to check as long as the time window for response is allowed.
> > > > >
> > > > > Paper #2887 Authors

---

> > > > > > ### Comment · Reviewer_GNhA · 2022-11-18
> > > > > > **On adjusting remix weights**
> > > > > >
> > > > > > Thanks again for the further clarifications. The ablation study does give us some insights.
> > > > > >
> > > > > > What is still unclear is how the proposed ED-HNN, which is a star expansion + equivariance approach, compares against the best easy-to-design (with all well-known tricks) clique expansion based approach without equivariance.
> > > > > >
> > > > > > In the discussion, the authors said
> > > > > > "ED-HNN outperforms GCN because it introduces new weights to remix central node features and neighborhood features. In simple graph regime, this amounts to the architecture adopted in [1][2]
> > > > > >
> > > > > > [1] Gilmer et al. Neural message passing for quantum chemistry, ICML 2017,
> > > > > >
> > > > > > [2] Chen et al. Simple and Deep Graph Convolutional Networks, ICML 2020"
> > > > > >
> > > > > > Based on this discussion, I think a strong baseline is a well-tuned GCNII on clique expansion equipped with all the well-known tricks (e.g., renormalisation, initial residual and identity mapping as in [2], presence/absence of self-loops/hypergraph representation whichever gives the better result).
> > > > > >
> > > > > > Please let me know if such a baseline is already compared against.

---

> > > > > > > ### Author Response · Authors · 2022-11-19
> > > > > > > **Response to Followup from Reviewer GNhA**
> > > > > > >
> > > > > > > Dear Reviewer GNhA,
> > > > > > >
> > > > > > > Thank you for notifying us of a new baseline CE+GCNII. We have added this experiment and the results are presented below. Per reviewer’s request, we adopted all the renormalization, initial+identity mapping and self-loops to GCNII.
> > > > > > >
> > > > > > > | Method | Cora | Citeseer | Pubmed | Cora-CA | DBLP-CA | Congress | Senate | Walmart | House |
> > > > > > > | ------------- | ------------- | ------------- | ------------- | ------------- | ------------- | ------------- | ------------- | ------------- | ------------- |
> > > > > > > | CE + GCNII | 78.43 $\pm$ 1.39 | 72.86 $\pm$ 1.74 | 86.59 $\pm$ 0.37 | 82.71 $\pm$ 1.50 | 91.36 $\pm$ 0.25 | 92.31 $\pm$ 0.79 | 49.06 $\pm$ 5.08 | 54.50 $\pm$ 0.53 | 65.08 $\pm$ 2.77 |
> > > > > > > | UniGCNII | 78.81 $\pm$ 1.05 | 73.05 $\pm$ 2.21 | 88.25 $\pm$ 0.40 | 83.60 $\pm$ 1.14 | 91.69 $\pm$ 0.19 | 94.81 $\pm$ 0.81 | 49.30 $\pm$ 4.25 | 54.45 $\pm$ 0.37 | 67.25 $\pm$ 2.57 |
> > > > > > > | ED-HNN  |80.31 $\pm$ 1.35 | 73.70 $\pm$ 1.38 | 89.03 $\pm$ 0.53 | 83.97 $\pm$ 1.55 | 91.90 $\pm$ 0.19 | 95.00 $\pm$ 0.99 | 64.79 $\pm$ 5.14 | 66.91 $\pm$ 0.41 | 72.45 $\pm$ 2.28 |
> > > > > > >
> > > > > > > We note that we have already compared with UniGCNII [1] in our original paper, which is an extension of GCNII to hypergraphs and serves as a strong baseline in our paper. As we show in our table, ED-HNN outperforms both UniGCNII and CE + GCNII on all the datasets. We think the reasons are as follows: 1) CE will actually lose more information than the bipartite graph representation of hypergraphs. So CE + GCNII performs worse than UniGCNII; 2) UniGCNII performs worse than ED-HNN as it does not have an equivariant architecture.
> > > > > > >
> > > > > > > [1] Huang et al. UniGNN: a Unified Framework for Graph and Hypergraph Neural Network
> > > > > > >
> > > > > > > Paper #2887 Authors

---

> > > > > > > > ### Comment · Reviewer_GNhA · 2022-11-19
> > > > > > > > **Concern on Sparse Hypergraphs Remain**
> > > > > > > >
> > > > > > > > Thanks again for the new experiments. The efforts of the authors during the response period are greatly appreciated! I still lean towards paper acceptance (albeit marginally) but now with an increased confidence score (4 -> 5).
> > > > > > > >
> > > > > > > > My initial concern that clique expansion based approaches might better process hypergraphs with a vast majority of size-two hyperedges, e.g., online forum thread networks, contact networks [1], still remains.
> > > > > > > >
> > > > > > > > During the response period, the authors conducted experiments on graph datasets (all hyperedges are of size two). On graph datasets, CE+GCNII reduces to GCNII. A quick comparison of ED-HNN performances on graph data (new table in the authors' response) and GCNII numbers on tables 2 and 5 in [2] gives a hint that clique expansion without equivariance might still be more effective on both sparse homophilic and sparse heterophilic datasets (i.e., vast majority of hyperedges are of size two).
> > > > > > > >
> > > > > > > > 1. [How Do Hyperedges Overlap in Real-World Hypergraphs? - Patterns, Measures, and Generators, In WebConf'21](https://dl.acm.org/doi/abs/10.1145/3442381.3450010)
> > > > > > > > 2. [Simple and Deep Graph Convolutional Networks, In ICML'20](https://proceedings.mlr.press/v119/chen20v.html)

---

> > > > > > > > > ### Author Response · Authors · 2022-11-19
> > > > > > > > > **Some other opinions on the Reviewer GNhA's concerns**
> > > > > > > > >
> > > > > > > > > Dear Reviewer GNhA,
> > > > > > > > >
> > > > > > > > > We thank the reviewer for appreciating our response. It is a pity that we are unable to ultimately match all of your expectation.
> > > > > > > > >
> > > > > > > > > First, we greatly appreciate reviewer GNhA's insightful questions and rigorous research attitude revealed during our discussion. In the same time, we respectfully disagree with reviewer GNhA's final concern and thus we would like to argue for our work. We do not argue for higher evaluation scores while we want to show the audience the full picture.
> > > > > > > > >
> > > > > > > > > > The concern from reviewer GNhA is that ED-HNN may fail to outperform clique expansion (CE) + GNN over hypergraphs whose hyperedges are mostly of size 2 and thus are very like graphs (named sparse hypergraphs).
> > > > > > > > >
> > > > > > > > > **Positive observation**: The effectiveness of the equivariant architecture can be shown via real-world hypergraph data: ED-HNN can outperform CE + GCNII over **hypergraphs even if more than half of the hyperedges are of size 2** based on the following table.
> > > > > > > > >
> > > > > > > > > | Method | Cora | Citeseer | Pubmed | Cora-CA | DBLP-CA | Congress | Senate | Walmart | House |
> > > > > > > > > | ------------- | ------------- | ------------- | ------------- | ------------- | ------------- | ------------- | ------------- | ------------- | ------------- |
> > > > > > > > > | CE + GCNII | 78.43 $\pm$ 1.39 | 72.86 $\pm$ 1.74 | 86.59 $\pm$ 0.37 | 82.71 $\pm$ 1.50 | 91.36 $\pm$ 0.25 | 92.31 $\pm$ 0.79 | 49.06 $\pm$ 5.08 | 54.50 $\pm$ 0.53 | 65.08 $\pm$ 2.77 |
> > > > > > > > > | ED-HNN  |80.31 $\pm$ 1.35 | 73.70 $\pm$ 1.38 | 89.03 $\pm$ 0.53 | 83.97 $\pm$ 1.55 | 91.90 $\pm$ 0.19 | 95.00 $\pm$ 0.99 | 64.79 $\pm$ 5.14 | 66.91 $\pm$ 0.41 | 72.45 $\pm$ 2.28 |
> > > > > > > > >
> > > > > > > > > **Negative observation**: Reviewer GNhA questions the effectiveness of the equivariant architecture because ED-HNN cannot outperform GCNII over **pure graphs** according to table 2 in [1] (table 5 is not directly comparable because of different data splitting ratios).
> > > > > > > > >
> > > > > > > > > We agree with these two observations. However, we respectfully disagree with the conclusion made by reviewer GNhA. Here are two reasons.
> > > > > > > > >
> > > > > > > > > 1. GCNII adopts invariant architecture + jump link. We argue that for the hyperedges of size 2, the jump link adopted by GCNII makes invariant architecture, i.e.,  invariant architecture + jump link $\rho'(h_v, \sum_{e:v\in e}\rho(\sum_{u\in e} \phi(h_u)))$, work very closely to equivariant architecture $\rho'(h_v, \sum_{e:v\in e}\rho(h_v, \sum_{u\in e} \phi(h_u)))$. We have explained the reason from the perspective of expressive power in the second last paragraph of Sec. 3.4. With some abuse of terminology, the gap between equivariant structure and invariant structure + jump link is "proportional" to the size $|e|$ from the perspective of expressive power. Therefore, for pure graphs, GCNII outperforming ED-HNN does not mean the failure of equivariant architecture. The performance difference is mainly due to other implementation details (most likely hyperparameter tuning or other non-essential reasons).
> > > > > > > > >
> > > > > > > > > 2. It is not convincing to use the failure of ED-HNN to outperform GCNII on "pure graphs" to argue that ED-HNN does not work over sparse hypergraphs. Because there is a missing definition of sparse hypergraphs. Basically, what ratio of hyperedges are of size 2, can we name it a sparse hypergraph? Moreover, even if ED-HNN cannot outperform CE+GCNII over hypergraphs say with at least 50% 2-size hyperedges, this does not reduce the value of ED-HNN because ED-HNN can outperform CE+GCNII over most real-world hypergraphs that do not lie in this regime as shown in the above table. This also does not mean the failure of our proposed equivariant architecture because of the reason 1. Note that this is a work for hypergraph models not simple graph models.
> > > > > > > > >
> > > > > > > > > Of course, the most crucial argument is that equivariant architecture + bipartite representation is essential in theory as it can represent hypergraph diffusion. This is nothing related to empirical performance or any implementation details.

---

> > > > > > > > > > ### Comment · Reviewer_GNhA · 2022-11-19
> > > > > > > > > > **On the Observations**
> > > > > > > > > >
> > > > > > > > > > Thanks for the observations.
> > > > > > > > > >
> > > > > > > > > > * While I agree that evaluation scores are not the be-all and end-all, the significance (to the hypergraph community) of a simple idea such as the one proposed in this paper (Step 3 in Algorithm 1) can be judged only when the experiments are conducted with utmost rigour.
> > > > > > > > > > * There is no fixed definition of sparsity but there definitely is still a lot of exploration (>50% - 100%) to be done between, say, co-citation networks (>50% hyperedges of size 2) and citation networks (100%).
> > > > > > > > > > * While I agree that equivariance, in general, is a very important idea, the main concern is to what extent the *combination of star expansion and equivariance* (which is the core idea of ED-HNN) competes with a clique expansion approach without explicitly incorporating equivariance. As the authors point out, invariance + a well-known trick such as jump link already works very closely to an equivariant architecture and so based on this reasoning, we could say a clique expansion + invariance + jump link approach implicitly works closely to an equivariant architecture on any hypergraph dataset.
> > > > > > > > > >
> > > > > > > > > > Points on experimental settings
> > > > > > > > > > * Important caveat for audience: If hyperparameter tuning or other non-essential reasons likely influence performance differences for negative observations, then they likely influence those for positive observations as well.
> > > > > > > > > > * To my knowledge, data splits are not standard on Cora, Citeseer, Pubmed in Table 5 of GCNII paper [2]. But data splitting is standard on heterophilic datasets in Table 5 and the same as those of a prior work [1]. If the authors have followed standard widely used splits [1], then they are directly comparable.
> > > > > > > > > >
> > > > > > > > > > [1] [Geom-GCN: Geometric Graph Convolutional Networks , In ICLR'20](https://openreview.net/forum?id=S1e2agrFvS)
> > > > > > > > > >
> > > > > > > > > > [2] [Simple and Deep Graph Convolutional Networks, In ICML'20](https://proceedings.mlr.press/v119/chen20v.html)

---

> > > > > > > > > > > ### Author Response · Authors · 2022-11-19
> > > > > > > > > > > **Greatly thank the reviewer while adding some further responses**
> > > > > > > > > > >
> > > > > > > > > > > We thank the reviewer a lot for all these comments. We greatly appreciate the reviewer's rigorous attitude again.
> > > > > > > > > > >
> > > > > > > > > > > We have to say that most of these conclusions are negotiable (from both their positive sides and their negative sides), especially on whether the regime with high sparsity should be explored to sufficiently justify the effectiveness of equivariance. In practice, it seems that only over citation networks (sparsity 100%), ED-HNN does not outperform GCNII while outperforming other more standard GNN baselines. Over common hypergraph benchmark datasets, ED-HNN always outperforms all other baselines. A philosophical question on this is: Do we really need to have a method outperform all current methods in the entire parameter space when claiming some superiority of that method? We think it is unnecessary as long as one gives the condition when the method may work better and such a condition covers a non-trivial range of practical problems. This is our understanding of doing research (perhaps, different people have different opinions). Also, we think most methods will not be the all winner in the entire parameter space (according to the hypothesis of no free lunch).
> > > > > > > > > > >
> > > > > > > > > > > Moreover, we would like to fix one incorrect argument made by the reviewer.
> > > > > > > > > > > The statement that **Invariance + jump link works similar to equivariance** is not correct. The right argument is that **Invariance + jump link works similar to equivariance when the hyperedge size is small** and the gap is proportional to the hyperedge size. This can be verified in theory.
> > > > > > > > > > >
> > > > > > > > > > > On the empirical side, Table 2 in the paper compares UniGCNII (invariance + jump link) with ED-HNN. Table 3 in the paper further compares HGNN/AllDeepSets + jump link (also invariance + jump link) with ED-HNN. In particular, here, we understand HGNN essentially adopts clique expansion. UniGCNII and AllDeepSets adopt star expansion. We also did ablation study during the response period by keeping all other parts unchanged and just moving from equivariance to invariance (note that here the latter has jump link):
> > > > > > > > > > >
> > > > > > > > > > > | Method | Cora | Citeseer | Pubmed | Cora-CA | DBLP-CA | Congress | Senate | Walmart | House |
> > > > > > > > > > > | ------------- | ------------- | ------------- | ------------- | ------------- | ------------- | ------------- | ------------- | ------------- | ------------- |
> > > > > > > > > > > | ED-HNN w/o Equiv. | 77.20 $\pm$ 1.15 | 71.65 $\pm$ 1.49 | 87.73 $\pm$ 0.64 | 81.91 $\pm$ 1.33  | 90.72 $\pm$ 0.27  | 87.52 $\pm$ 1.49 | 48.25 $\pm$ 4.87 | 62.80 $\pm$ 0.63 | 65.40 $\pm$ 2.59 |
> > > > > > > > > > > | ED-HNN  |80.31 $\pm$ 1.35 | 73.70 $\pm$ 1.38 | 89.03 $\pm$ 0.53 | 83.97 $\pm$ 1.55 | 91.90 $\pm$ 0.19 | 95.00 $\pm$ 0.99 | 64.79 $\pm$ 5.14 | 66.91 $\pm$ 0.41 | 72.45 $\pm$ 2.28 |
> > > > > > > > > > >
> > > > > > > > > > > So, we have many experiments that test empirical differences on invariance + jump link v.s. equivariance, :)

---

> > > > > > > > > > > > ### Comment · Reviewer_GNhA · 2022-11-19
> > > > > > > > > > > > **On Whether High Sparsity Should be Explored**
> > > > > > > > > > > >
> > > > > > > > > > > > Thanks again for discussion. While I agree that ED-HNN is demonstrated to be effective on existing hypergraph node classification benchmarks, what made the difference was the authors claiming that "ED-HNN further shows great superiority in processing heterophilic hypergraphs."
> > > > > > > > > > > >
> > > > > > > > > > > > Heterophilic hypergraphs are a very general concept which includes a *huge space of datasets*. In homophilic data, we tend to see nodes belonging to the same label in a hyperedge, but in heterophilic data, we do not need to satisfy any such condition. Hence, the space of datasets to explore is really huge in heterophilic case and, relatively speaking, considerably limited in homophilic case.
> > > > > > > > > > > >
> > > > > > > > > > > > It remains unclear how effective ED-HNN would be compared to CE + invariance + jump link when the homophily level is very low (much less than 50%). Additionally, node classification problems in sparser hypergraphs such as contact networks (e.g., predict gender) and online forum thread networks (e.g., predict thread creation time) are very likely going to be heterophilic node classification tasks. The space of datasets covered in the paper is limited if the authors claim superiority on heterophilic hypergraphs (without stating any specific conditions on heterophily).
> > > > > > > > > > > >
> > > > > > > > > > > > Finally, there are a lot of selling points (superiority on heterophily, equivariance, simple yet significant) for a small change (step 3 in algorithm 1). There is definitely a need for rigorous empirical evaluation to demonstrate the significance of the the small change.

---

> > > > > > > > > > > > > ### Author Response · Authors · 2022-11-19
> > > > > > > > > > > > > **Thank you so much for the comments**
> > > > > > > > > > > > >
> > > > > > > > > > > > > We would like to thank the reviewer again for the time to discuss with us. We have learned a lot from this discussion.
> > > > > > > > > > > > >
> > > > > > > > > > > > > We have some results already and follow-up questions for the comments the reviewer added.
> > > > > > > > > > > > >
> > > > > > > > > > > > > **1. It remains unclear how effective ED-HNN would be compared to CE + invariance + jump link when the homophily level is very low (much less than 50%).**
> > > > > > > > > > > > >
> > > > > > > > > > > > > We have such results in Table 3 in the paper. We use $\alpha$ to control different levels of homophily. When $\alpha = 7$, it means 7 nodes in the hyperedge of size 15 (used in the experiments) come from one class and the rest 8 nodes come from the other class. This is already the most heterophilic case. Also, the baseline HGNN + JumpLink adopts CE+invariance+jump link. We also have four real-world heterophilic datasets with homophilic coefficients close to 0.5.
> > > > > > > > > > > > >
> > > > > > > > > > > > > **2. node classification problems in sparser hypergraphs such as contact networks (e.g., predict gender) and online forum thread networks (e.g., predict thread creation time) are very likely going to be heterophilic node classification tasks.**
> > > > > > > > > > > > >
> > > > > > > > > > > > > We guess that the reviewer referred to the datasets in paper [1]. We checked the datasets in the github of [1] while we did not see the node labels. Does the reviewer know about where to find such node labels? We can do the experiments on the final version of this paper if it gets accepted.
> > > > > > > > > > > > >
> > > > > > > > > > > > > Actually, when we did this work, we also tried hard to find extremely heterophilic hypergraphs from real-world datasets. We have already adopted several very heterophilic hypergraphs we could find in the paper. We have not seen a real-world hypergraph with homophilic coefficients substantially less than 0.5. If the reviewer knows more those kinds of datasets, we are happy to look into them.
> > > > > > > > > > > > >
> > > > > > > > > > > > > [1] Lee et al., How Do Hyperedges Overlap in Real-World Hypergraphs? - Patterns, Measures, and Generators

---

> > > > > > > > > > > > > > ### Comment · Reviewer_GNhA · 2022-11-19
> > > > > > > > > > > > > > **Challenges in Heterophilic Hypergraphs**
> > > > > > > > > > > > > >
> > > > > > > > > > > > > > The synthetic experiments referred to in the discussion are very restricted. For example, they are not real-world, restricted to only two different classes in a hyperedge (local), and restricted to binary classification (global).
> > > > > > > > > > > > > >
> > > > > > > > > > > > > > Real-world datasets are not readily available but can be curated from their original sources.
> > > > > > > > > > > > > >
> > > > > > > > > > > > > > For example, contact network data can be obtained from prior work [1]
> > > > > > > > > > > > > >
> > > > > > > > > > > > > > [1] [Contact Patterns in a High School: A Comparison between Data Collected Using Wearable Sensors, Contact Diaries and Friendship Surveys](https://www.ncbi.nlm.nih.gov/pmc/articles/PMC4556655/)

---

> > > > > > > > > > > > > > > ### Author Response · Authors · 2022-11-19
> > > > > > > > > > > > > > > **Regarding the heterophilic hypergraph datasets**
> > > > > > > > > > > > > > >
> > > > > > > > > > > > > > > We thank the reviewer a lot for pointing out the reference. Since the real-world datasets are not readily available (this is also one of the reasons we adopted synthetic datasets), we guess it may take us some time to build new datasets. We will try in our final version.
> > > > > > > > > > > > > > >
> > > > > > > > > > > > > > > For the contact network by using gender as node labels, it will be most likely binary classification (perhaps LGBT labels are so few). We guess it is still restrict based on the reviewer's argument. We were wondering if the reviewer can point out more datasets, specifically including which the reviewer will think our study is sufficient. We greatly appreciate it.

---

> > > > > > > > > > > > > > > > ### Comment · Reviewer_GNhA · 2022-11-19
> > > > > > > > > > > > > > > > **On node labels**
> > > > > > > > > > > > > > > >
> > > > > > > > > > > > > > > > Gender is not the only possible label for a node. Properties like Age and Class are also potential labels. If a student attends multiple classes then it can also become a multi-label classification problem.

---

> > > > > > > > > > > > > > > > > ### Author Response · Authors · 2022-11-19
> > > > > > > > > > > > > > > > > **Thanks!**
> > > > > > > > > > > > > > > > >
> > > > > > > > > > > > > > > > > We will try to look into that hypergraph and the metadata the reviewer refers to. Many thanks!

---

### Official Review · Reviewer_n5MN · 2022-10-24

**Confidence:** 2
**Correctness:** 2
**Technical Novelty And Significance:** 2
**Empirical Novelty And Significance:** 3
**Recommendation:** 6

**Clarity, Quality, Novelty And Reproducibility:**

Clarity

The paper is well-written. I had no great difficulty understanding the main points of the paper.


Quality

I want to clarify the relationship between ED-HNN and the corresponding hypergraph diffusion problem. Certainly, Proposition 3 showed that ED-HNN could approximate any hypergraph diffusion operator. However, the opposite direction is not known, i.e., whether any ED-HNN represents a hypergraph diffusion operator. If we think of ED-HNN as a hypergraph diffusion operator, there should exist functions $f$ and $g$ such that the model represents an optimization algorithm of $\min \sum_v f(h_v; x_v) + \sum_e g(H_e)$. However, given ED-HNN, we do not know (at least explicitly) such functions.
It is true that in Section 4.4, this paper showed that ED-HNN could numerically approximate the functions (CE, TV, LEC) used for the objective function of hypergraph diffusion. However, this only demonstrates the approximation capability of ED-HNN and does not imply that ED-HNN is a hypergraph diffusion operation.
For these reasons, although I understand there is an analogy between hypergraph diffusion and ED-HNN, hypergraph diffusion does not work as a justification for ED-HNN.


Novelty

If my understanding is correct, extensions of the representation theorem (Zaheer et al., 2017) for invariant functions to the equivariant case appeared in existing studies (for example, [Sannai et al., 2019, Corollary 3.2]). If that would be true, I would say the novelty of Theorem 1 is limited.

[Sannai et al., 2019]: https://arxiv.org/abs/1903.01939


Reproducibility

The Appendix describes how to prepare the dataset used in the experiment and the hyperparameters of the prediction model. No code is provided, so there is no guarantee of perfect reproduction. However, I think we can implement the code to reproduce the experiments to some degree.

**Details Of Ethics Concerns:**

N.A.

**Strength And Weaknesses:**

Strengths
- Numerical evaluation used various types of graph data. It demonstrates the broad applicability of the proposed model.

Weaknesses
- The parameterization of NNs somewhat hinders the relationship between the proposed method and underlying hypergraph diffusion (see the Quality section for details).
- Similar results exist for the representation theorem for equivariant functions (Theorem 1) in existing studies.

**Summary Of The Paper:**

This paper proposed ED-HNN, a GNN for hypergraphs inspired by the algorithm for solving hypergraph diffusion (optimization problem on hypergraphs). ED-HNN approximates the gradient-based optimization algorithm for hypergraph diffusion by a message passing on a bipartite graph that is equivalent to the original hypergraph. As a theoretical analysis, this paper showed the representation theorem for equivariant functions and justified the architectural design of ED-HNN. Finally, this paper applied ED-HNN to various graph data as an empirical evaluation.

**Summary Of The Review:**

The proposed method shows good prediction accuracy in numerical experiments. On the other hand, on the theoretical side, the proposed method is justified by associating it with hypergraph diffusion. However, I have questions about the validity of the arguments and would like to clarify them. Also, if I understand correctly, the representation theorem for equivariant functions, one of this paper's contributions, is known in existing studies.

# Post-rebuttal comments

After the discussion with the authors, I increase my score (5 -> 6). (Note that there is a possibility that my final evaluation could change after the further discussion with other reviewers and area chairs)

I raised two weak points in the first review comment. One is that the relationship between the hypergraph diffusion problem and the proposed model is unclear. The other is that the universality result (Theorem 1) is known. The question related to the first question was solved through the discussion with the authors. I understand that the proposed model is a gradient-based optimization of a hypergraph diffusion problem for some unknown function $f$ and $g$. Also, making $m^{(t)}_{e\to v}$ equivariant, which is inspired by the hypergraph diffusion, certainly improved the empirical performance.
On the other hand, my concerns about the second question were correct. However, the second point was not strong compared with the first point.
In conclusion, since I recognize the novelty and significance, I increase my score from 5 to 6, tending to accept.

---

> ### Author Response · Authors · 2022-11-14
> **Response to Reviewer n5MN**
>
> Dear Reviewer n5MN,
>
> We thank reviewer n5MN for the time and effort reviewing our paper, as well as the appreciation of our experiments. First of all, we appologize that the attached source code zip is not compatible with some operating system. We have fixed this problem and updated with a new copy. Reviewer n5MN remains concerned about our theoretical contribution. Overall, the connection between hypergraph diffusion and HNNs is not missing by using NN. Also, our Theorem 1 differs from prior works. Please see our detailed responses are as below:
>
> **1. Whether any ED-HNN represents a hypergraph diffusion operator?**
>
> The ED-HNN indeed represents the hypergraph diffusion operator because it aligns with the gradient descent of the original potential functions (see our Appendix C). That being said, taking the integral of ED-HNN will give the energy function $f$ and $g$. However, such $f$ and $g$ may not have the closed form (because even though derivative has a closed form and is integrable, the closed form of its original function may not exist). Moreover, an optimal choice of $f$ and $g$ is often unknown in practice. However, we can avoid their specific closed forms by approximating their derivatives with neural networks.  ED-HNN shares the same flavor of algorithm unrolling methods such as [1, 2, 3], which unfold iterative (optimization) algorithms and train a neural network to compute each iteration. In our scenario, ED-HNN is trained to reconstruct the original hypergraph diffusion problem by learning the gradient from data. Empirically, we compared our model with HyperND which has a pre-defined $f$ and $g$. Our performance is advantageous over HyperND suggests a data-driven diffusion process can generalize better than fixed diffusion .
>
> [1] Gregor et al. Learning fast approximations of sparse coding. ICML 2010.
>
> [2] Zhang et al. Learning deep cnn denoiser prior for image restoration. CVPR 2017.
>
> [3] Song et al. Generative Modeling by Estimating Gradients of the Data Distribution, NeurIPS 2019.
>
> **2. The representation theorem for equivariant functions, one of this paper's contributions, is known in existing studies.**
>
> We clarify that our core contribution lies in building the theoretical connection between HNN and hypergraph diffusion. Theorem 1 is not the only theoretical contribution. We note that the summation in [Sannai et al., 2019, Corollary 3.2] pools over elements *other than the central node*, while our method sums over *all the elements*. Although this looks like a tiny difference, the proof technique and implementation are totally different. [Sannai et al., 2019] is proposed to model set functions but not for hypergraphs with coupled sets. To implement [Sannai et al., 2019] in message passing, one has to maintain each node-hyperedge pair (E x N). Each time, the node-hyperedge pair have to save the node features together with the features aggregated from other nodes within a hyperedge, which is computationally prohibitive and less practical to implement. While in ED-HNN, we only need to maintain a group of node features and a group of hyperedge features separately (E + N). Each time hyperedge pools over all connected node features, and then each node extracts information from connected hyperedge features.
>
> Best,
>
> Paper #2887 Authors

---

> > ### Comment · Reviewer_n5MN · 2022-11-17
> > **Response to authors' comments**
> >
> > I thank the authors for replying to my comment.
> >
> > **1. Whether any ED-HNN represents a hypergraph diffusion operator?**
> >
> > If I understand correctly, the authors claim that the one-layer of ED-HNN represents the approximation of the right-hand side of (2) for the unknown function $f$ and $g$. However, I am not sure this is an appropriate justification. I agree that there exists some (time-dependent) function $f^{(t)}$ and $g^{(t)}$ such that (2) holds by replacing $f$ with $f^{(t)}$ (resp. $g$ with $g^{(t)}$). However, the claim above is stronger in that $f^{(t)}$'s are the same and independent of $t$. I am not sure the differences among $f^{(t)}$'s are small enough to regard them as errors derived from approximating the right-hand side of (2) with ED-HNN.
> >
> >
> > **2. The representation theorem for equivariant functions, one of this paper's contributions, is known in existing studies.**
> >
> > > We note that the summation in [Sannai et al., 2019, Corollary 3.2] pools over elements other than the central node, while our method sums over all the elements.
> >
> > If my understanding is correct, we can absorb this difference by appropriately modifying $\rho$ and $\phi$. Specifically, given $\rho(z_i, \sum_{j} \phi(z_j))$, by setting $\rho'(u, v) = \rho (u, v) + \phi(u)$, we have $\rho'(z_i, \sum_{j\not =i} \phi(z_j)) = \rho(z_i, \sum_{j} \phi(z_j))$.
> > Conversely, given $\rho'(z_i, \sum_{j\not = i} \phi(z_j))$ by setting $\rho(u, v)= \rho'(u, v) - \phi(u)$, we have $\rho(z_i, \sum_{j} \phi(z_j)) = \rho'(z_i, \sum_{j\not =i} \phi(z_j))$.
> >
> > Therefore, I think Theorem 1 in this paper and [Corollrary 3.2 of Sannai et al., 2019] are equivalent.
> >
> > Regarding implementation, I agree that using $\rho(z_i, \sum_{j} \phi(z_j))$ is efficiently implementable, assuming that the architecture of $\rho$ and $\rho'$ are almost the same.

---

> > > ### Author Response · Authors · 2022-11-17
> > > **Response to Followup from Reviewer n5MN**
> > >
> > > Dear Reviewer n5MN,
> > >
> > > Thank you for initiating an insightful discussion on the theoretical side of our work. Please see our responses as below:
> > >
> > > **1. Whether any ED-HNN represents a hypergraph diffusion operator?**
> > >
> > > We appreciate reviewer n5MN for raising this insightful question. However, actually, both the ground-truth diffusion operators and our modeled operators are all time-invariant, as long as the step size is not changed during the optimization. To see this, please check Eq. 2 and Eq. 3,4, all gradient operators $\nabla f$, $\nabla g$, proximal operators $prox_{f}$, $prox_{g}$ are all time-invariant. The time-variant parts are the inputs of these operators, i.e., $X^{(t)}$ and $H^{(t)}$. Because those ground-truth operators are time-invariant, we adopt time-invariant (i.e., layer-invariant) models that share a neural network across all the layers. So, there is no concern about mismatch between time-variant and time-invariant parts
> > >
> > > **2. The representation theorem for equivariant functions, one of this paper's contributions, is known in existing studies.**
> > >
> > > We appreciate the reviewer's insightful derivations, which actually coincide with our proof technique. Technically, the proof of Theorem 1 is not hard given the result in [Sannai et al., 2019] (We thank the reviewer again for reminding us of this result. We have acknowledged this result in the proof). However, this does not reduce much the main contributions made by this paper, because the primary goal of Theorem 1 is to guide the efficient implementation of neural networks that universally represent hypergraph diffusions, where the complexity of the induced implementation is definitely an important factor to consider. The observation of the connection between hypergraph diffusion and the right neural network architectures is the key theoretical finding of this work. The specific content in Theorem 1, as we have noted in the paper, is an extra contribution (say claim on Page 5).
> > >
> > > Paper #2887 Authors

---

> > > > ### Comment · Reviewer_n5MN · 2022-11-18
> > > > **t is an optimization step index rather than layer index**
> > > >
> > > > I thank the authors for answering my questions. And I am sorry that my comments about Q1 were different from what I intended.
> > > >
> > > > Specifically, my concern was that the approximation target $\nabla f$ and $\nabla g$ could change as we train the model because learnable parameters changes during training.
> > > > That is, in the previous comment, I intended to use $t$ in $f^{(t)}$ and $g^{(t)}$ as an optimization step index (e.g., the number of minibatches processed during training), rather than layer index of the model.
> > > >
> > > > Regarding the authors' response, I agree that $f$ and $g$ do not depend on the layer index.
> > > >
> > > > I am sorry for the confusion.

---

> > > > > ### Author Response · Authors · 2022-11-19
> > > > > **Response to Followup from Reviewer n5MN**
> > > > >
> > > > > Dear Reviewer n5MN,
> > > > >
> > > > > Many thanks for your response and clarification. There is no such problem that the reviewer is worried about. To avoid confusion, next, we use $\rho_k, \phi_k$ to denote the ED-GNN models after $k$-step training. We keep using $t$ to denote the layer index to avoid further confusion.
> > > > >
> > > > > **Our short answer:** There is no such problem that the reviewer is worried about. Because the learned $f$ and $g$ are irrelevant to the training dynamics of  $\rho_k, \phi_k$, i.e. how $\rho_k, \phi_k$ evolves w.r.t. $k$. The learned $f$ and $g$ are only related to the convergent point of the training, i.e., $(\rho_{\infty}, \phi_{\infty}) = \lim_{k\rightarrow \infty} (\rho_k, \phi_k)$. Once the model converges, the composition of $\rho_{\infty}$ and $\phi_{\infty}$ by following the form of Eq. 5 actually gives the learned $\nabla f$, $\nabla g$ (or $prox_f$, $prox_g$), which further has their corresponding functions $f$ and $g$. Basically, only the convergent point of the training indicates the learned diffusion instead of the intermediate steps during the training. You may also think of the procedure in the way that if a machine learning problem is to learn a function $f$ by using a parameterized model $\hat{f}$. $\hat{f}$ will get trained $\hat{f_k}$, k=1,2,3,.... What we only care about is the convergent point of the training $\lim_{k\rightarrow\infty} \hat{f_k}$, which determines the final model we learn and the performance.
> > > > >
> > > > > To avoid further confusion, note that the iteration procedure of Eq. 2 does not correspond to the training procedure of $\rho, \phi$ but corresponds to the layer iteration in the model, i.e., the iteration t=1,2,...,L-1 in Algorithm 1.
> > > > >
> > > > > **Our relatively long answer:** If we understand it correctly, we see the confusion of reviewer n5MN come from some misunderstandings of the framework of algorithm unrolling [1]. Algorithm unrolling has found applications in many domains, especially deep learning for signal and image processing [2-4].  We explain the logic of algorithm unrolling in our setting as follows.
> > > > >
> > > > > > The classical way to obtain hypergraph diffusion: People handcraft different types of potential functions $f$ and $g$ for specific tasks (e.g., TV (w/ p=2)  for node denoising by assuming Gaussian noises, and more examples are raised in Example 1). Then, based on these handcrafted potential functions, solve the hypergraph diffusion Eq. (1) and the solution can be used for inference tasks over hypergraphs.
> > > > >
> > > > > The issue with this classical way is that the optimal potential functions $f$ and $g$ that best fit the inference tasks over hypergraphs are often unknown. So, this motivates our neural-based formulation to learn data-driven $f$ and $g$.
> > > > >
> > > > > > ED-HNN: Suppose there are some unknown potential functions f and g to be learned. Of course, $\nabla f$, $\nabla g$ (or $prox_f$, $prox_g$) are also unknown. We use parameterized architectures $\rho, \phi$ to model $\nabla f$, $\nabla g$ (or $prox_f$, $prox_g$). Then, we use data from the inference tasks (e.g., node classification) to train $\rho, \phi$. We denote the training procedure as $(\rho_{\infty}, \phi_{\infty}) = \lim_{k\rightarrow \infty} (\rho_k, \phi_k)$. Then, the convergent $(\rho_{\infty}, \phi_{\infty})$ corresponds to our learned $\nabla f$, $\nabla g$ (or $prox_f$, $prox_g$) based on ED-HNN. Note that only the convergent point $(\rho_{\infty}, \phi_{\infty})$ of the training matters, while $(\rho_k, \phi_k)$ during the training is not relevant.
> > > > >
> > > > > [1] Monga et al. Algorithm Unrolling: Interpretable, Efficient Deep Learning for Signal and Image Processing
> > > > >
> > > > > [2] Gregor et al. Learning fast approximations of sparse coding. ICML 2010.
> > > > >
> > > > > [3] Zhang et al. Learning deep cnn denoiser prior for image restoration. CVPR 2017.
> > > > >
> > > > > [4] Song et al. Generative Modeling by Estimating Gradients of the Data Distribution, NeurIPS 2019.
> > > > >
> > > > > Paper #2887 Authors

---

> > > > > ### Author Response · Authors · 2022-11-20
> > > > > **Response to the follow-up question "t is an optimization step"**
> > > > >
> > > > > We were wondering if we have addressed the reviewer's question on what $f$ and $g$ are finally learned. We are looking forward to your further questions if there are any.
> > > > >
> > > > > Many thanks!
> > > > > Authors

---

> > > > > ### Author Response · Authors · 2022-11-23
> > > > > **A further comment on when the training does not converge to a point but converges to a point with fluctuation**
> > > > >
> > > > > Dear Reviewer n5MN,
> > > > >
> > > > > Since we have not heard back from reviewer n5MN, we suspect reviewer n5MN may be still confused by our argument on the equivalence between ED-HNN and hypergraph diffusion. Below we provide more discussion on how ED-HNN could approximate any hypergraph diffusion even if the training procedure does not converge to a point but contains some fluctuations of the model parameters.
> > > > >
> > > > > Formally, we denote the learned gradient by ED-HNN as $\nabla f_k$ (and also $\nabla g_k$) that is given by $\rho_k$ and $\psi_k$, where $k$ is the training step. When $k$ is large, we consider the most practical scenario: Even if the model does entirely converge ($\nabla \hat{f}_k$ contains some fluctuation), the model at least will converge to some points near a stationary point that $\nabla f_k = \nabla f + \epsilon_k$ where $\epsilon_k$ is a small fluctuation. Suppose $||\cdot||$ is L2 norm. Actually, as long as
> > > > >
> > > > > (a) $||\epsilon_k||<\epsilon$ is bounded,
> > > > > (b) $f_k$ and $f$ are defined on some compact convex space $\mathcal{X}$,
> > > > >
> > > > >
> > > > > the maximal error between $f$ and $f_k$ up to a constant, $\min_{c} \sup_{x\in \mathcal{X}} \|f(x) - f_k(x) + c\|$,  will be upper bounded by $\epsilon$ times the diameter of $\mathcal{X}$.  Actually, the constant difference $c$ comes from the integrating procedure but it is irrelevant because it will not impact the final diffusion results according to Eq. (1) (adding a constant to $f$ or $g$ does not affect the obtained $h$).
> > > > >
> > > > > $\min_{c} \sup_{x\in \mathcal{X}} \|f(x) - f\_k(x) + c\| $
> > > > >
> > > > > $= \min_{c} \sup_{x\in \mathcal{X}} \| \int_s \nabla f(x) - \int_s  \nabla f\_k(x) + f(x_0) -  f\_k(x_0) + c\| $
> > > > >
> > > > > $\leq  \min_{c} \sup_{x\in \mathcal{X}}(\| \int_s \nabla f(x)dx - \int_s  \nabla f_k(x) dx\| + \|f(x_0) -  f\_k(x\_0) + c\|) $
> > > > >
> > > > > $= \sup_{x\in \mathcal{X}}\| \int_s \nabla f(x)dx - \int_s  \nabla f\_k(x)dx\| $
> > > > > $\leq \sup_{x\in \mathcal{X}}\int_s || \nabla f(x) - \nabla f\_k(x)||\cdot||dx||  \leq
> > > > > \epsilon \cdot \text{diag}(\mathcal{X}).$
> > > > >
> > > > > where $s$ is the shortest path (in norm $||\cdot||$) from $x_0$ to $x$.  Therefore, as long as $\epsilon$ is small, then $f_k$ is a good estimation of $f$ in a compact space with an L2 path to $x_0$.

---

> > > > > > ### Comment · Reviewer_n5MN · 2022-11-27
> > > > > > **Sorry for the late response**
> > > > > >
> > > > > > Dear Authors
> > > > > >
> > > > > > I appreciate the authors that gave me further explanations. I am sorry that I have not responsed for a long time. I will reconsider my concern and reply to the comments soon.
> > > > > >
> > > > > > Best,
> > > > > > Reviewer n5MN

---

> > > > > > ### Comment · Reviewer_n5MN · 2022-11-30
> > > > > > **I understand**
> > > > > >
> > > > > > Again I thank the authors for sincerely answering my concerns. I think I understand the authors' explanation, and my concern is solved.

---

> > > > > > > ### Author Response · Authors · 2022-11-30
> > > > > > > **Thanks! Would you please re-evaluate our work?**
> > > > > > >
> > > > > > > Dear reviewer n5wn,
> > > > > > >
> > > > > > > Many thanks for checking our response! We greatly appreciate your time and consideration. We were wondering if you may kindly re-evaluate our work since we have addressed your concerns. We guess the only open question is that a partial result of Thm 1 can be proved based on a previous paper. However, such overlap does not decrease much of this work's main contribution. We have explained that such a partial result cannot establish the connection between the equivariant hypergraph NN models and hypergraph diffusion approaches, which can only be done by our analysis. Moreover, we are the first ones who try to establish such a connection in the hypergraph/graph learning community. We have given some credit to the previous result. Given this, we are looking forward to reviewer n5wn's reevaluation.
> > > > > > >
> > > > > > > Many thanks,
> > > > > > > the authors

---

> > > > > > > > ### Comment · Reviewer_n5MN · 2022-12-02
> > > > > > > > **I raise my score**
> > > > > > > >
> > > > > > > > After the discussion with the authors, I increase my score (5 -> 6). (Note that there is a possibility that my final evaluation could change after the further discussion with other reviewers and area chairs)
> > > > > > > >
> > > > > > > > I raised two weak points in the first review comment. One is that the relationship between the hypergraph diffusion problem and the proposed model is unclear. The other is that the universality result (Theorem 1) is known. The question related to the first question was solved through the discussion with the authors. I understand that the proposed model is a gradient-based optimization of a hypergraph diffusion problem for some unknown function $f$ and $g$. Also, making $m^{(t)}_{e\to v}$ equivariant, which is inspired by the hypergraph diffusion, certainly improved the empirical performance.
> > > > > > > > On the other hand, my concerns about the second question were correct. However, the second point was not strong compared with the first point.
> > > > > > > > In conclusion, since I recognize the novelty and significance, I increase my score from 5 to 6, tending to accept.

---

> ### Author Response · Authors · 2022-12-02
> **Thanks for your re-evaluation!**
>
> Dear reviewer n5MN,
>
> We greatly thank your time for re-evaluating our work!
>
> We would like to state our understandings on the second concern so as to provide more information to the final discussion. The previous work [Sannai et al., 2019, Corollary 3.2] does not give the exact form of Theorem 1. Please check the diagram 2 in the paper [Sannai et al., 2019]. They still use $[...,\rho(h_v, \sum_{u\neq v} \phi(h_u)),...]$ to keep equivariance instead of $[...,\rho(h_v, \sum_{u} \phi(h_u)),...]$ in this work. First, one still needs an extra step of derivation to get Theorem 1, though we agree that the step is mathematically simple. Second, this small change is significant for hypergraph diffusion representation, because it means we do not need to manually construct pseudo nodes to represent node-hyperedge pairs to get universal approximation. The construction of those pseudo nodes, adopted by a few previous works will substantially increase the computation/memory complexity.
>
> [Sannai et al., 2019] https://arxiv.org/pdf/1903.01939.pdf

---

### Official Review · Reviewer_z25x · 2022-10-25

**Confidence:** 2
**Correctness:** 3
**Technical Novelty And Significance:** 3
**Empirical Novelty And Significance:** 3
**Recommendation:** 6

**Clarity, Quality, Novelty And Reproducibility:**

The presentation is good.
Reproducibility: N/A.


**Details Of Ethics Concerns:**

None.

**Strength And Weaknesses:**

**Strength:**
 - 1. The presentation is clear.
 - 2. The overall experiments show the superiority of their proposed algorithm, ED-HNN.

**Weakness:**

The main claims of the paper are not well explained or supported. In particular, section 3.3 tried to discuss the advantages of ED-HNN for the design of equivariant diffusion operators. However, I have several questions in this section.
- 1. For the claim that “Invariant diffusion by forcing the operation on hyperedge e to follow that the partial derivation to each dimension is the same”, could you please refer to some reference or give proof on this? It is not obvious for a reader that is not familiar to the invariant diffusion,
- 2. “Moreover, a learnable operator is important.” This conclusion is weak, since it only tells that it exists such parameters that can help but didn’t say whether it is difficult to optimize to such parameters. Could you give some empirical experiments on the sensitivity of the parameters to show whether the learnable operator is difficult to optimize?
- 3. “equivariant hyperedge diffusion operators are also good at building deep models” is not well supported and explained. Although Figure 2 shows an interesting result that ED-HNN successfully leverages deeper architecture to achieve higher accuracy, there is no clear trend to show that How about the performance of ED-GNN when the number of the layers go to larger than 8? Meanwhile, the claim that “Equivariant operators allocating different messages across nodes helps with overcoming the oversmoothing issue.” is not obvious to me. I could see those equivalent operators may reduce the risk of overfitting, but how do equivalent operators help the over-smoothing issue? Is there any insight on this point?

**Summary Of The Paper:**

This paper proposes a hyper-graph neural network model, ED-HNN, that can model hypergraph diffusion process. They claimed that ED-HNN shows superiority in processing heterophilic hypergraphs and constructing deep models.

**Summary Of The Review:**

Please refer to the Strength And Weaknesses. If the authors could address the problems in the weakness, I’d like to raise my scores.

---

> ### Author Response · Authors · 2022-11-14
> **Response to Reviewer z25x**
>
> Dear Reviewer z25x
>
> We thank reviewer z25x for appreciating our empirical results and giving a positive initial assessment. The code provided in original supplementary material cannot be opened in some system. Hence, we have re-uploaded a compatible version. The main concerns of Reviewer z25x lie in the statements in Sec. 3.3. We provide point-by-point responses to your questions below:
>
> **1. Explanation for “Invariant diffusion by forcing the operation on hyperedge e to follow that the partial derivation to each dimension is the same”.**
>
> Invariant diffusion corresponds to a permutation invariant function (see Def. 4) whose outputs are unchanged if only permuting the input order. The mathematical definition of an invariant diffusion is: for any node feature matrix $X$, the output of invariant diffusion is $1 \psi(X)^\top$ where $\psi: \mathbb{R}^{N \times F} \rightarrow \mathbb{R}^F$ is an invariant function. In contrast to equivariant diffusion, the invariant diffusion operator summarizes the node information to one feature vector and passes this identical message to all nodes uniformly. In the language of message passing, invariant diffusion usually first pools node features to edge $e$ by an invariant function $h_e = \psi(\{x_j | j \in e\})$ and then the edge $e$ identically dispatches this message to all connected nodes $x_i = \sum_{e : i \in e} h_e$. To be more clear, we add a figure and section in our **Appendix D** to elaborate this comparison. Concrete instances of invariant models include HyperGCN [1] and HGNN [2]. In all of our experiments (Tab. 2 and Tab. 3), the invariant models lack expressiveness and behave poorly on heterophilic datasets.
>
> [1] Yadati et al. Hypergcn: A new method for training graph convolutional networks on hypergraphs
>
> [2] Feng et al. Hypergraph neural networks
>
> **2. “Moreover, a learnable operator is important”.This conclusion is weak, since it only tells that it exists such parameters that can help but didn’t say whether it is difficult to optimize to such parameters**
>
> We appreciate the reviewer raising this insightful question. We leave discussing the optimization properties of ED-HNN for future work. On the empirical side, we trained an ED-HNN on both realistic and synthetic heterophilic hypergraph datasets as well as to fit a given diffusion process. The results are present in Tab. 2, Tab. 3, and Fig. 3, respectively. The superior performance indicates common gradient-based optimizers (e.g., Adam) have no difficulty to optimize our ED-HNN. Our ED-HNN outperforms a fixed hypergraph diffusion algorithm HyperND, which validates the benefit of a learnable diffusion. Here we only focus on discussing the expressiveness of HNN (i.e., universal approximation power) in handling heterophilic data. The example reveals a fixed diffusion is not enough while a learnable diffusion can bring more flexibility and fit the heterophilic labels better. In fact, our ED-HNN layers are implemented by neural networks, which are amenable to being optimized end-to-end.
>
>
> **3. Equivariant hyperedge diffusion operators are also good at building deep models” is not well supported and explained. How do equivalent operators help the over-smoothing issue?**
>
> Intuitively, over-smoothing is caused because neighborhood information dominates the node features, making features within a connected component indistinguishable. ED-HNN can also circumvent over-smoothing by our equivariant operator $\rho(h_u, \sum_v \phi(h_v))$ because it passes distinctive messages to different nodes, which makes the node features less likely to be the same. The remapped message in (Algorithm 1, Step 3) is computed by involving the central node feature. Such an operation can preserve the local features as it plays a similar effect to the identity connection in GNN literature [1]. **Moreover, we added experiments with 16-layer & 32-layer models in Fig. 2.** Our model still performs well when scaling up to 32 layers while other models suffer from performance degradation.
>
> [1] Chen et al. Simple and Deep Graph Convolutional Networks, ICML 2020
>
> Best,
>
> Paper #2887 Authors

---

> > ### Comment · Reviewer_z25x · 2022-11-21
> > **Thank you.**
> >
> > Thank you for your detailed explanations. I am still positive about the acceptance of this paper. But the score remains the same.

---

> > > ### Author Response · Authors · 2022-11-21
> > > **Thanks!**
> > >
> > > We greatly thank reviewer z25x for checking our response! We think our responses have addressed the questions about the weaknesses the reviewer mentioned.  If more things need to be addressed, please let us know.
> > >
> > > Authors

---

### Official Review · Reviewer_TN8f · 2022-10-25

**Confidence:** 3
**Correctness:** 2
**Technical Novelty And Significance:** 3
**Empirical Novelty And Significance:** 3
**Recommendation:** 6

**Clarity, Quality, Novelty And Reproducibility:**

The clarity is OK, but it would be better if the high-level idea of the model can be explained more clearly. In the current version, the complex theories and formulations make it hard to gain an overview of how the model works.

Quality is OK.

Novelty is good.

Not sure about the reproducibiliy. The model is sufficiently described in paper, but whether the results can be reproduced can only be chekced by running the code.

**Strength And Weaknesses:**

Strengths:

This paper has a clear motivation on the proposed model.

The proposed model is accompanie with theoretical analysis.

The experiments are comprehensive. Mltiple datasets and baselines are included.

Weakness:

1. The adopted datasets seem not to be hypergraph datasets. As introduced in the paper, in a hypergraph, a hyperedge connects a set of nodes. However, in datasets such as cora, citeseer, pubmed, each edge only connect two nodes, and they are just standard graph datasets. Isn't there any dataset that contain true hyper edges？It is not convincing that the simple dataset like cora requires many higher-order relations to capture during classification.

2. Since some of the adopted datasets are standard graph datasets, which were also widely adopted in works on standard GNNs, like GCN [1], the compared baselines should also include these methods. As far as I know, the performance of GCN on Cora is higher than any method shown in Table 2.

[1] Kipf, Thomas N., and Max Welling. "Semi-supervised classification with graph convolutional networks." arXiv preprint arXiv:1609.02907 (2016).

**Summary Of The Paper:**

This paper studies the learning on hyper graphs. Targeting the problem that existing hyper graph networks often require domain knowledge in designing the mechanism to process the higher-order relations in hyper graphs, this paper proposes a hypergraph diffusion based model that can model a wide range of higher-order relations without explicitly designing based on domain knowledge. Theoretical analysis is also provided on the proposed model. Expeiments are conducted on 9 datasets and are compared with 8 baselines.


**Summary Of The Review:**

Overall, this paper has a clear motivation targetng a meaningful problem, which is the good aspect. However, there is also some unlcear part regarding the experiments. Therefore, I would rate it as slightly below threshold currently

---

> ### Author Response · Authors · 2022-11-14
> **Response to Reviewer TN8f**
>
> Dear Reviewer TN8f,
>
> We thank reviewer TN8f for reviewing our paper and acknowledging our theoretical depth. First of all, we hope to clarify that we had already attached our reproducible code in our initial submission. However, the zip package seems not compatible with different platforms. We have fixed this problem and updated the supplementary material. We are under the impression that reviewer TN8f may misunderstand our experiment setting. To be more clear, we have added a detailed specification to our datasets, and our responses are as below:
>
> **1. The adopted datasets seem not to be hypergraph datasets, so standard GCNs should be compared.**
>
> These three citation networks we used in our paper are no more simple graphs but hypergraphs. For co-citation networks (Cora, Citeseer, Pubmed), all documents cited by a document are connected by a hyperedge. For co-authorship networks (Cora-CA, DBLP), all documents co-authored by an author are in one hyperedge.  See Sec. 12 in [2] for more details. These three datasets have served as a common benchmark to evaluate HNN models, e.g.,[1,2]. We have also included a detailed specification of these datasets in our revision. In this sense, comparing with GCN baselines is beyond our scope since they work on the tasks of different domains.
>
> [1] Eli Chien et al. You are allset: A multiset function framework for hypergraph neural networks, ICLR 2022.
>
> [2] Yadati et al. HyperGCN: A New Method of Training Graph Convolutional Networks on Hypergraphs, NeurIPS 2019
>
> **2. It would be better if the high-level idea of the model can be explained more clearly.**
>
> A high-level way to understand ED-HNN is to view ED-HNN as a hypergraph-diffusion-inspired hypergraph neural network. By revisiting the hypergraph diffusion, we notice that although hypergraph diffusion and HNNs have similar message passing paradigm, equivariance is missing in modern HNNs. Motivated by this, we proposed our theoretically grounded architectural change (Step 3 in Algorithm 1) to improve the expressiveness of HNNs by provably achieving the equivariance.
>
> Though our rigorous arguments in the paper may look obscure, the principle and our proposed techniques are quite simple. The key principle is to pick up the missing equivariance in HNN and the key design is Step 3 in Algorithm 1. Upon existing message passing in HNNs, Step 3 additionally interacts the target node feature with the aggregated neighborhood features before dispatching to the target node. This change is neat and crucial to implement expressive equivariant operators.
>
> Best,
>
> Paper #2887 Authors

---

> > ### Comment · Reviewer_TN8f · 2022-11-19
> > **post-rebuttal responses**
> >
> > Thanks for the explanations from the authors. My concern is resolved and I will increase the score

---

> > > ### Author Response · Authors · 2022-11-21
> > > **Thanks!**
> > >
> > > We greatly thank reviewer TN8f to check our response and reevaluate our work!
> > >
> > > Authors

---

### Decision · Program_Chairs · 2023-01-20

**Decision:**

Accept: poster

**Justification For Why Not Higher Score:**

The paper seems not a significant contribution to the ML community, although it contains certain value.

**Justification For Why Not Lower Score:**

The paper is nicely written, with good theoretical analysis and comprehensive experimental evaluations.

**Metareview: Summary, Strengths And Weaknesses:**

This paper studies the learning on hyper graphs, to tackle the challenges that existing approaches often require domain knowledge in designing the mechanism to process the higher-order relations in hyper graphs. Theoretical analysis is provided on the proposed model and experiments are conducted on quite a few datasets with many baselines.

Overall speaking, this paper has done a nice job. It receives consistent feedback from all the reviewers. They believe that this paper has both merits and limitations. For example, the paper has a clear motivation, nice theoretical analysis, and comprehensive experiments. On the other hand, some of the claims in the paper were not very well supported, and the choices of the datasets need more justifications. There is a long thread of discussions between the authors and our reviewers. I can see that the authors have made great efforts on revising their paper and carefully explaining to the reviewers. I believe the value of the paper becomes clearer through these discussions and it should be fine to accept it to ICLR.


**Note From Pc:**

if the above contains the word "oral" or "spotlight" please see: "oral" presentation means -> notable-top-5% and "spotlight" means -> notable-top-25%. As stated in our emails, we are disassociating presentation type from AC recommendations